

# Regional-scale Hydrologic Model Comparison Including Calibration for Improved River Discharge Simulations into the Mediterranean Sea

Mohamed Hamitouche[1,2,3], Giorgia Fosser[1], Arezoo RafieeiNasab[4], Alessandro Anav[2,3]

[1]University School for Advanced Studies IUSS, Pavia, Italy
[2]Climate Modeling Laboratory, ENEA – Italian National Agency for New Technologies, Energy and Sustainable Economic Development, CR Casaccia, Viale Anguillarese 301, 00123 Santa Maria di Galeria, Rome, Italy
[3]ICSC Italian Research Center on High-Performance Computing, Big Data and Quantum Computing, Bologna, Italy
[4]Research Applications Laboratory, NSF National Center for Atmospheric Research, Boulder, Colorado, USA

*Correspondence to*: Mohamed Hamitouche (mohamed.hamitouche@iusspavia.it)

**Abstract.** River discharge into the Mediterranean Sea is a vital component of the regional water cycle, influencing ecological and climatic dynamics. Although some regional coupled models that include a river routing component exist for the Mediterranean region, their performance in reproducing river discharge is poor. This study compares the hydrological routing models CaMa-Flood and WRF-Hydro for discharge simulations into the Mediterranean Sea. Evaluating their performance across key basins, this study highlights CaMa-Flood's computational efficiency but underperformance in flow variability and high-flow extremes, contrasted by WRF-Hydro's superior timing and bias reduction, especially after calibration. In fact, results indicate that the calibration improved WRF-Hydro's metrics, including Kling-Gupta Efficiency (KGE) and lag times, underscoring its potential for precise discharge predictions at higher computational costs. These findings offer critical insights for advancing regional coupled Earth system models, enhancing hydrological forecasting, and addressing basin-specific hydrological challenges.

## 1 Introduction

River discharge into the Mediterranean Sea is a critical component of its water budget, alongside the net inflow from both the Atlantic through the Strait of Gibraltar and the Black Sea via the Dardanelles Strait, as well as evaporation and precipitation (Pinardi et al., 2015; Pinardi and Masetti, 2000). As one of the primary freshwater sources, river discharge plays a pivotal role in shaping the basin's hydrological and ecological dynamics. It does not only provide essential freshwater input, particularly during spring when plentiful precipitation rates and snowmelt enhance the discharge, but also transports nutrients and minerals that influence coastal and sub-basin ecosystems (Struglia et al., 2004). Moreover, variability in river discharge, whether natural or anthropogenic, can modulate the Mediterranean's thermohaline circulation on decadal scales, affecting salinity, dense water formation, and oxygenation rates across the basin (Zavatarelli et al., 1998).





As climate change, population growth, and human interventions increasingly impact river systems, the need for timely and accurate river discharge estimates becomes ever more critical for water resource management, flood prediction, and risk mitigation (Nearing et al., 2024). In this context, understanding the interplay between regional climate change and hydrological processes is essential, particularly as extreme hydrological events such as floods and droughts become more frequent and severe. These interactions underscore the need to study the coupling of Earth system components, including

atmospheric, hydrological, and oceanic processes. Since the early 21st century, major international research programs such as WCRP (World Climate Research Programme), IGBP (International Geosphere-Biosphere Program), GEWEX (The Global Energy and Water Cycle Experiment) and CORDEX (COordinated Regional climate Downscaling EXperiment) have emphasized the importance of integrating regional climate models (RCMs) with hydrology and ocean models to address these challenges. Coupled atmospheric-hydrological-ocean models provide a framework to simulate the full water cycle and

its feedback mechanisms between land, atmosphere and ocean. Such models reveal how terrestrial water flows influence the broader water cycle and regional climate dynamics over seasonal to decadal scales. For example, terrestrial water flow can alter atmospheric boundary layers and modulate convective precipitation during shorter time scales (Amelia, 2022). By enhancing the representation of water cycle processes, these coupled models aim to improve simulation accuracy and forecasting capabilities. This evolution from traditional hydrological models to coupled models reflects the growing need for

tools that capture the intricate interactions between land, ocean, and atmosphere, offering enhanced weather forecasts and improved predictions of river flow and extreme events.

Regional coupled models and climate change studies in the Mediterranean require a high-resolution discharge component to correctly reproduce the complex orography and land-sea distribution of the Mediterranean region and thus effectively close the coupling among the Earth system components (Hagemann et al., 2020). However, the first coupled models did not

consider the hydrological component and the coupling was limited to atmosphere-ocean-land only (e.g. RegIPSL (Shahi et al., 2022), EBU-POM (Djurdjevic and Rajkovic, 2010) and MORCE (Drobinski et al., 2012)), where rivers are inadequately represented. This restricts their ability to make accurate predictions of river flow and forecasts for floods and droughts, necessitating downstream hydrological modelling, independently of land-atmosphere feedback or the advantages of data assimilation. Additionally, while some models incorporate atmosphere-ocean-land-river coupling, they struggle to meet the

high-resolution requirements essential for precise discharge simulations.

Nowadays, several complex coupled models, developed to achieve fully integrated hydrological predictions, exist for the Mediterranean region. Despite the sophistication to simulate complex earth system interactions, these models still face significant challenges. In particular, some models systematically underestimate freshwater input from river runoff, leading to inaccuracies in discharge predictions and contributing to surface water salinification in the Mediterranean (Anav et al., 2021;

Reale et al., 2020; Storto et al., 2023). This underestimation seems to be with the HD model (Hagemann and Dümenil, 1997) used to simulate the river discharge with different horizontal resolutions (e.g. 5 minutes in MESMAR and 0.5 degrees in ENEA-REG). The HD model uses a pre-parametrization based on a linear reservoir routing concept with pre-defined reservoir numbers and temporal constants tailored to runoff inputs from the HydroPy LSM (Stacke and Hagemann, 2021),





which neglects the energy budget and overestimates runoff. Consequently, replacing the biased routing model in these
coupled systems has become essential to ensure a more accurate representation of the water cycle.

At the same time, enhancing discharge simulations would also benefit from selecting the most appropriate land surface
runoff model based on its runoff generation mechanism. A recent study by Hamitouche et al. (2025) analysed the impact of
seven different runoff schemes within the Noah-MP LSM on global discharge simulations across diverse climate regions and
found that the Schaake runoff scheme performed best in warm temperate regions, including the Mediterranean basins. This
study utilized the CaMa-Flood hydrodynamic model (Yamazaki et al., 2011) for discharge simulation, demonstrating overall
good performance against observational discharges. However, the analysis also revealed certain limitations, such as delays in
capturing seasonal peak flows due to inherent constraints in CaMa-Flood, which were accurately resolved by the WRF-
Hydro model (Gochis et al., 2021), tested in the same study. Additionally, the study identified significant biases, particularly
in high-flow extremes, emphasizing the need for ongoing calibration of tuneable parameters to improve the accuracy of
hydrological predictions.

This highlights the importance of conducting detailed sensitivity analyses on standalone routing models, before adding them
into coupled climate or Earth system models.

Thus, considering the poor performance in reproducing accurate discharge estimates as well as some limitations of the
existing regional coupled model for the Mediterranean region, the aim of this study is to compare the performances of two
process-based hydrological routing models, i.e. CaMa-Flood and WRF-Hydro, in reproducing the discharge for the most
important Mediterranean rivers. This evaluation provides valuable information, as the analysed models could be regarded as
alternatives to the river component used in current regional coupled models. CaMa-Flood, a global river routing model, is
widely recognized for its computational efficiency and ability to simulate river discharge at large scales. However, within the
Med-CORDEX domain, it has been utilized only outside the context of regional coupled models. On the other hand, WRF-
Hydro, the hydrological extension of the WRF atmospheric model, is designed for high-resolution hydrological predictions,
with multi-scale capabilities, enabling it to represent processes on various spatial scales (Gochis et al., 2021). Over the
Mediterranean region, its application has primarily been limited to small isolated and relatively undisturbed basins, and short
time periods (Galanaki et al., 2021; Senatore et al., 2015; Sofokleous et al., 2023, 2024). In contrast, in this study,
simulations were conducted at a daily time scale over a long-term period (1990-2014) for the entire Med-CORDEX domain.
The evaluation focused on several Mediterranean basins as well as the Danube River, which drains into the Black Sea,
allowing for robust regional generalizations and comparative analyses across diverse hydrological regimes. Additionally, the
study analyzed the role of parameter calibration in improving discharge simulations, with a particular focus on WRF-Hydro,
leveraging the capabilities of the NCAR WRF-Hydro calibration package.

This study addresses the following key questions:

•    Can WRF-Hydro or CaMa-Flood serve as effective alternatives to improve hydrological simulations within Euro-
         Mediterranean regional coupled models?

      •    Can calibration enhance WRF-Hydro performance, and to what extent?



The paper is structured as follows: after the presentation of the used models and methods (section 2), the first part of result (section 3.1) focuses on comparing the two hydrological models in their default configurations, interpreted as a baseline reference, offering insights into the foundational performance of these models. This approach establishes a basis for comparison and lays the groundwork for future studies to refine and adapt these models for innovative applications, particularly in Mediterranean hydrological and climatic contexts. The second part of the results (section 3.2) evaluates the impact of calibration in further improving discharge simulations, highlighting the potential of parameter optimization to enhance hydrological model performance, with a particular focus on WRF-Hydro, leveraging the capabilities of the NCAR WRF-Hydro calibration package.

## 2 Materials and methods

### 2.1 Study area and river discharge observations

The river basins draining into the Mediterranean Sea encompass over 5 million km², including the Nile basin, but excluding rivers flowing into the Atlantic Ocean from Portugal and Spain (Lionello et al., 2012; Ludwig et al., 2009). Most of these catchments are medium to small-scale, with only a few major basins exceeding 80,000 km² (Lionello et al., 2012). The ten largest rivers contributing to Mediterranean discharge include Rhone, Po, Drin-Buna, Nile, Neretva, Ebro, Tiber, Adige, Seyhan and Ceyhan rivers (Ludwig et al., 2009), with 71% of the total discharge originating from northern Mediterranean countries, 12% from eastern regions (Turkey), and 17% from southern areas, primarily the Nile. Notably, the Rhone and the Po alone contribute 25% of the northern discharge. Annual freshwater input to the Mediterranean and Black Sea is estimated at 305–737 km³/year (Struglia et al., 2004).

Considering the ENEA-REG model (Anav et al., 2021), the ocean component incorporates the discharge, simulated by the river routing model, from 18 main rivers, including Rhone, Po, Ebro, Ceyhan, Adige, Tiber, Drin, Meric, Goeksu, Vjosa, Jucar, Buyuk Menderes, Arno, Kopru, and Struma, as boundary conditions. For the Nile, a climatological monthly mean is prescribed, as suggested by the Med-Cordex protocol (https://zenodo.org/records/11659642), due to the atmospheric model's limited domain coverage of the basin and the significant anthropogenic modifications to its natural discharge.

In this study, the validation focuses on 10 key rivers flowing into the Mediterranean Sea, in addition to the Danube River (Fig. 1), for which at least five consecutive years of daily observations were available for validation. These rivers have different climatic and morphologic conditions, ranging from mountainous alpine regions with pluvio-nival hydrological regimes to the semi-arid climate of southern Turkey's Ceyhan River. Other river basins were excluded due to insufficient daily observational data post-1990 or data spanning less than five years. The Júcar and Nile rivers were specifically excluded because their flow is heavily influenced by human interventions, such as reservoirs and water diversions, which are not explicitly represented in the modelling setup. For each selected river, simulated discharge was compared to observations from the nearest available station to the river mouth, covering upstream areas between 1,900 and 95,000 km² (807,000 km² for the Danube). Observational data were sourced from the Global Runoff Data Centre (GRDC, 2024), the Hydrographic



Studies Center of CEDEX (CEH-CEDEX, 2024), and from Hagemann et al. (2020). To ensure accurate hydrograph construction and validation metrics, simulated discharge values corresponding to missing observational data were removed.

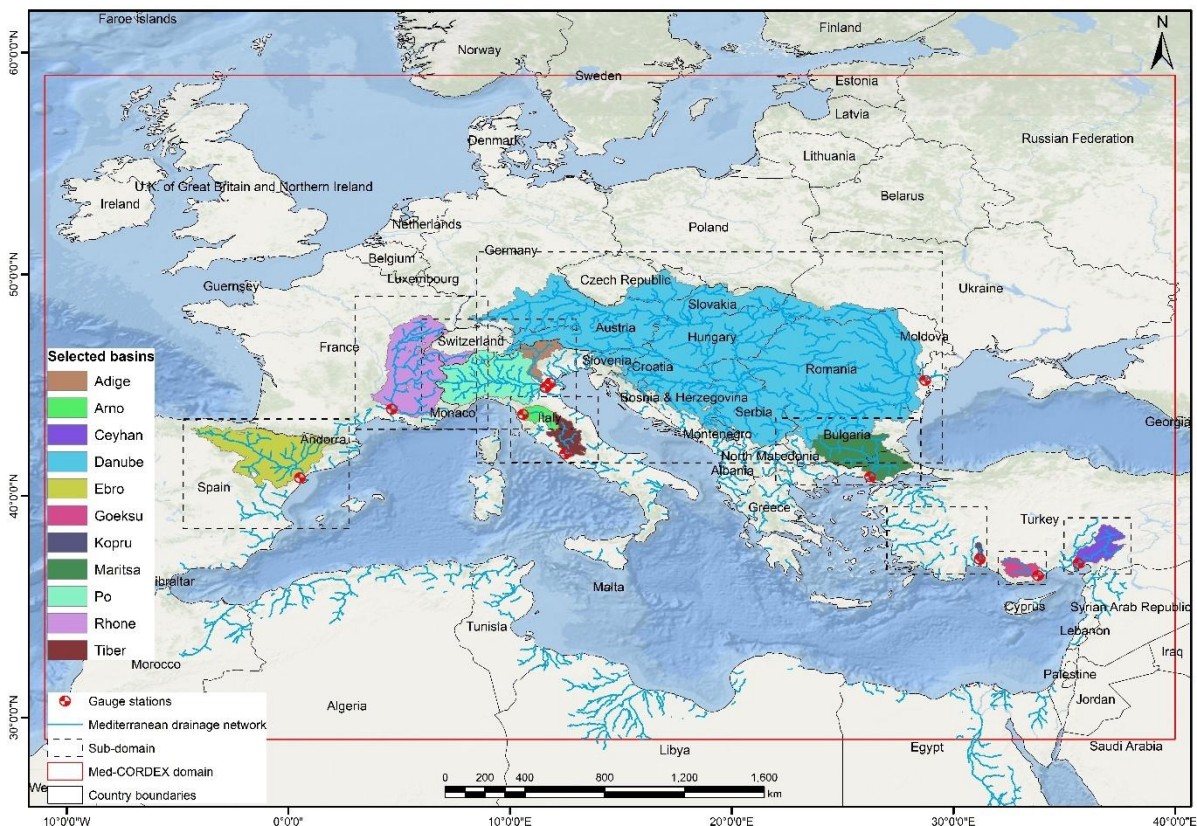

**Figure 1: Med-CORDEX simulation domain with its drainage network and the gauge stations (red dots) used in this study. The selected basins for evaluation and calibration are given distinct colours. The dashed contours refer to calibration sub-domains.**

**2.2 Model description and experimental setup**

This study is based on the use of CaMa-Flood and WRF-Hydro hydrological models for daily discharge simulations, driven by the ENEA-REG atmosphere-land-ocean coupled model run at 12km resolution over the Med-Cordex domain. Within ENEA-REG, WRF model is used to dynamically downscale ERA5 data (Hersbach et al., 2020) and simulate the atmospheric variables, while the Noah-MP model simulates the runoff. The configurations of both models (WRF and Noah-MP) are

provided in Table S1 and Table S2 in the supplement. All the ENEA-REG variables required as input by the two hydrological models are previously regridded to 6 km resolution using conservative remapping. In particular, CaMa-Flood uses as input daily runoff (surface + subsurface runoff, Fig. S1 in the supplement), while WRF-Hydro, which couples Noah-MP with the hydrological routing model, requires a set of atmospheric state variables —such as air temperature, surface pressure, specific humidity, horizontal wind components (10 m), downward shortwave and longwave radiation, and rainfall

rate— provided at 6-hourly intervals (Fig. S2 in the supplement).



Both CaMa-Flood and WRF-Hydro hydrological models are run for the period 1990-2014, after five years of spin-up. None of the simulations considers reservoir operations and lakes.

### 2.2.1 CaMa-Flood processes and configuration

CaMa-Flood (v4.11, Yamazaki et al., 2011, 2013) is a global-scale distributed hydrodynamics model designed to simulate river discharge, water levels, and floodplain inundation by routing runoff from land surface models through a predefined river network map. The model employs a Manning's friction coefficient of 0.03 for main river channels and automatically adjusts its routing time step to satisfy the Courant-Friedrichs-Lewy condition as used in previous studies (e.g., Bates et al., 2010; Yamazaki et al., 2013).

CaMa-Flood employs the local inertial equation (a simplified form of the Saint-Venant equation that excludes the advection 155 term) to balance computational efficiency with physical accuracy. River basins are represented as unit catchments with subgrid parameters for floodplain topography, allowing floodplain inundation to be modelled as a subgrid-scale process. River discharge is calculated between upstream and downstream catchments based on the grid-vector hybrid river network, while water levels and inundation areas are diagnosed from water storage in each catchment, assuming uniform water surface elevation. Water storage is updated by conserving mass, incorporating upstream inflows, downstream outflows, and 160 local runoff inputs. Floodplains of neighbouring unit catchments can exchange flows through so-called bifurcation channels, making it a quasi-2D model.

The river network map and subgrid topography parameters were extracted using the FLOW upscaling algorithm (Yamazaki et al., 2009), applied to MERIT Hydro hydrography maps (Yamazaki et al., 2019) and MERIT DEM data at 3 arcsec (90 m) resolution. Errors and obstacles in the DEMs, such as vegetation canopy, levees, water surface contamination, and radar 165 speckles, were corrected to ensure consistent downhill flow along streamlines (Yamazaki et al., 2012).

### 2.2.2 WRF-Hydro processes and configuration

WRF-Hydro (v5.2, Gochis et al., 2021) is a hydrological model designed to simulate surface and subsurface runoff processes, either as a standalone model as in our case or coupled with the WRF atmosphere model. It integrates overland flow, subsurface flow, baseflow, and channel routing to represent river discharge and hydrodynamics. WRF-Hydro solves 170 the Boussinesq equation for saturated subsurface lateral flow, accounting for hydraulic gradients driven by topography, saturated soil depth, and hydraulic conductivity. It incorporates exfiltration from fully saturated cells and infiltration excess from the integrated Noah-MP land surface model (LSM), ensuring a dynamic interaction between surface and subsurface processes that together contribute to the 2-D overland flow and the 1-D channel routing.

Overland and channel flows are computed using the diffusive wave equation, an efficient approximation of the Saint-Venant 175 equations that captures backwater effects and shallow water dynamics. The current surface water representation assumes one-way water flow into channels, without simulating overbank flow. Additionally, baseflow is represented using a conceptual storage-discharge bucket model, which gradually returns water to downstream channels.



Both the land surface model (LSM) and the hydrological routing model components of WRF-Hydro were run on the same 6 km spatial resolution grid. The land surface simulations were conducted at hourly intervals, while discharge was routed

using a conservative terrain and channel routing timestep of 6 minutes. Streamflow outputs were recorded at a daily frequency.

For a fair comparison, WRF-Hydro was run using the same topography generated for CaMa-Flood. However, because WRF-Hydro requires high-resolution topography to generate the drainage network through its GIS preprocessing tool, the upscaled DEM was reconditioned both automatically and manually. This reconditioning was based on an available validated drainage

network at a close resolution of 1/16° (Wu et al., 2012) and involved minimal intervention to avoid significantly altering flow velocity.

### 2.2.3 WRF-Hydro Calibration

Hydrological models, whether physical or conceptual, require calibration to achieve reliable streamflow simulations. This necessity arises from two key challenges: (i) the inability to measure all model parameters at the scale of application, and (ii)

the simplification and spatiotemporal discretization of the complex and highly variable rainfall-runoff processes (Beck et al., 2020).

The WRF-Hydro modelling system includes various predefined hydrological parameters, some of which are influenced by land use (e.g., vegetation type and density) and soil type (e.g., silt, clay, loam, sand) (Cerbelaud et al., 2022; Verma and J., 2023). These parameters can be adjusted or calibrated to account for the specific regional orographic and climatic

characteristics (Lahmers et al., 2019; Yu et al., 2023), such as those of the Mediterranean basin, where default parameter values are often insufficient. Calibration involves determining optimal tuneable coefficients by proportionally adjusting these spatial parameters relative to their default values, ultimately improving simulation accuracy (Cerbelaud et al., 2022; Verma and J., 2023). The WRF-Hydro calibration parameters are categorized into six groups related to: soil properties, runoff processes, groundwater dynamics, vegetation behaviour, snowpack melting, and channel characteristics. In this study,

calibration focused on the first four groups, covering 16 parameters (Table S3), while snowmelt parameters were left at their default settings due to the limited relevance of snowmelt in most of the study region and for potential future regionalization over other snow-free basins. Similarly, channel parameters, such as channel roughness, were not calibrated because the channel routing resolution of ~6 km was considered too coarse for a reliable hydrological calibration.

For the calibration of the WRF-Hydro model, the Dynamically Dimensioned Search (DDS) algorithm, a heuristic single-

solution-based global optimization method developed by Tolson and Shoemaker (2007), was employed by running the NCAR's (National Centre for Atmospheric Research) PyWrfHydroCalib package (https://github.com/NCAR/PyWrfHydroCalib). This algorithm is particularly suited for complex, high-dimensional hydrological models, enabling efficient exploration of parameter space while minimizing computational demands. DDS begins with a global search that transitions to a more localized search as iterations progress, guided by a scalar

neighbourhood size perturbation parameter, which is typically set to 0.2. The dynamic adjustment of search size and




probabilistic selection of parameters ensure rapid convergence toward good local or regional global optima, even without explicitly targeting the true global optimum (Tolson and Shoemaker, 2007). The stopping criterion is the number of user-defined iterations, which was set to 350 in this study. The calibration objective function, the Kling-Gupta Efficiency (KGE) (Gupta et al., 2009), is optimized to maximize agreement between simulated and observed streamflow. This method has been

shown to yield efficient solutions for WRF-Hydro (Cosgrove et al., 2024; RafieeiNasab et al., 2025), balancing computational efficiency with calibration accuracy.

Additionally, to address the computational challenges posed by the WRF-Hydro model's complexity and the grid-based structure, which results in longer simulation times compared to other hydrological models (Kiliçarslan, 2022), a sub-domain approach was adopted to optimize calibration efficiency. Instead of calibrating the hydrological basins across the entire Med-

CORDEX domain, smaller sub-domains (Fig. 1) were created. Each sub-domain focused on one or two basins (when their calibration date ranges overlapped) and was designed to closely match the Med-CORDEX grid, albeit with minor deviations. Calibration was then performed independently for each basin using an HPC system to ensure efficient utilization of computational resources. On average, the calibration simulations required a bit less than 3000 CPU hours per basin, effectively balancing performance and scalability to address the varying sizes and computational demands of the river

basins. Following calibration, the optimized parameters for each basin were reintegrated into the larger Med-CORDEX domain for validation.

Before calibration, WRF-Hydro was "spun-up" for five years using default model parameters for each selected basin. This spin-up phase was critical for stabilizing soil moisture initial conditions. The model state at the end of this spin-up period served as a "warm start" for the calibration phase, which was performed for five years for each basin. Each calibration

iteration included a one-year spin-up to align the model state with current conditions and suppress instabilities from parameter changes. The specific spin-up and calibration periods varied across basins and were selected based on the availability of reliable and continuous observational records, i.e. at least 5-year all falling within the 1990–2014 period. The optimized parameters from the calibration were then used to evaluate the model over the entire 1990–2014 period, focusing on ensuring consistent performance across the selected basins. Station-observed streamflow data served as the reference for

both calibration and validation, providing a robust basis for model evaluation. Figure S3 in the supplement summarizes the different calibration steps.

## 2.3 Metrics

The evaluation of the performance of the different hydrological models relied on several complementary metrics to capture various aspects of hydrological behaviour, including correlation, bias, and flow variability. The metrics used in this study are

described below:

1. Spearman's Rho (Spearman, 1904):

This non-parametric measure of rank correlation assesses the monotonic relationship between simulated and observed streamflow, providing insight into the consistency of flow ranking without assuming linearity. Spearman's rank-order





correlation coefficient, used as an indication of flow timing, was chosen instead of Pearson's coefficient due to its

effectiveness in handling nonlinear and non-normally distributed data, such as hydrologic time series (Yue et al., 2002).

2. Kling-Gupta Efficiency (KGE) (Gupta et al., 2009):

The KGE was used as a comprehensive measure of model performance, combining correlation, bias, and variability.

3. Relative Standard Deviation (rSD):

The rSD compares the variability in simulated and observed streamflow, expressed as a ratio between the standard deviations

of the simulated and observed streamflow.

4. Percent Bias (%Bias): The %Bias quantifies the systematic error in the simulation as a percentage of observed values. This metric, highlights over- or underestimation of the total flow by the model.

5. Low-flow Percent Bias (Bottom 30%) (Casper et al., 2012):

This metric evaluates the model's ability to simulate long-term low-flow conditions by calculating the bias for the bottom

30% of observed streamflow values, which are critical for drought analysis and ecological studies.

6. High-flow Percent Bias (Top 2%) (Yilmaz et al., 2008):

To assess the model's performance in simulating peak discharge events, the bias for the top 2% of observed streamflow values was computed. This is particularly important for understanding flood dynamics and extreme events.

7. Time Lag:

The lag corresponding to the highest cross-correlation between observed and simulated streamflow was calculated to assess the timing discrepancies. This metric helps identify whether the model captures the correct temporal alignment of flow dynamics with observations.

Together, these metrics provide a comprehensive evaluation framework, addressing different aspects of hydrological performance, from overall state to extreme conditions. Their corresponding value ranges and equations are detailed in Table

265 1.

**Table 1: List of evaluation statistical metrics. In particular, d is the difference in independent ranking for simulated and observed values for day i and n is the number of values in each time series. $\sigma_{sim}$ and $\sigma_{obs}$ are the standard deviations of the simulated and observed streamflow, respectively. $QO_i$ and $QS_i$ are observed and simulated flow values for day i, respectively. r is the Pearson correlation coefficient, $\beta$ is the relative bias (mean simulated divided by mean observed), and $\gamma$ is the variability ratio (standard**
**deviation of simulated divided by standard deviation of observed). $QO_l$ and $QS_l$ are the bottom 30% observed and simulated flow values. $Q_{min}$ is the minimum flow value in the observed and simulated timeseries. $\mu_{obs}$ and $\mu_{sim}$ are the means of observed and simulated flow values.**

| Metric | Description | Range (Perfect) | Equation | |
|---|---|---|---|---|
| $r_s$ | Spearman's rho | -1 to 1 (1) | $$r_s = 1 - \frac{6 \sum d_i^2}{n \times (n^2 - 1)}$$ | (1) |
| rSD | Relative Standard | 0 to Inf (1) | $$rSD = \frac{\sigma_{sim}}{\sigma_{obs}}$$ | (2) |



| | Deviation | | |
|---|---|---|---|
| %Bias | Percent Bias | -100 to Inf (0) | $$\%Bias = 100 \times \frac{\sum_{i=1}^{N}(QS_i - QO_i)}{\sum_{i=1}^{N} QO_i} \qquad (3)$$ |
| KGE | Kling-Gupta Efficiency | -Inf to 1 (1) | $$KGE = 1 - \sqrt{(r-1)^2 + (\beta - 1)^2 + (\gamma - 1)^2} \qquad (4)$$ |
| %BiasFLV | Low-flow Percent Bias | -100 to Inf (0) | $$\%BiasFLV = 100 \times \left( \frac{\sum_{l=1}^{L} log\left(\frac{QS_l}{Q_{min}}\right)}{\sum_{l=1}^{L} log\left(\frac{QO_l}{Q_{min}}\right)} - 1 \right) \qquad (5)$$ |
| %BiasFHV | High-flow Percent Bias | -100 to Inf (0) | $$\%BiasFHV = 100 \times \frac{\sum_{h=1}^{N}(QS_h - QO_h)}{\sum_{h=1}^{N} QO_h} \qquad (6)$$ |
| $\tau$ | Time lag | >= 0 (0) | Corresponds to the maximum value of the cross-correlation $r_\tau$: $$r_\tau = \frac{\sum_{i=1}^{n-\tau}(QO_i - \mu_{obs}) \times (QS_{i+\tau} - \mu_{sim})}{\sqrt{\sum_{i=1}^{n-\tau}(QO_i - \mu_{obs})^2 \sum_{i=1}^{n-\tau}(QS_{i+\tau} - \mu_{sim})^2}} \qquad (7)$$ |

## 3 Results and discussion

### 3.1 WRF-Hydro vs. CaMa-Flood (default configurations)

In this section, we compare the performance of CaMa-Flood and WRF-Hydro in its default configuration, while a detailed discussion of the calibrated WRF-Hydro is provided in the next section. To evaluate the predictive performance of the models, we used measures of accuracy, precision, and discrimination—percent bias, standard deviation, and Spearman's rho, respectively. These metrics are presented in a Taylor diagram (Fig. 2). Lin et al. (2019) and Sanchez Lozano et al. (2025) indicate that a good bias and variability should be within the range of ±20%, representing values for the bias and variability

ratio from 0.8 to 1.2. For Spearman's rho, Tijerina-Kreuzer et al. (2021) used a threshold greater than 0.5 as an indicator of a good shape.





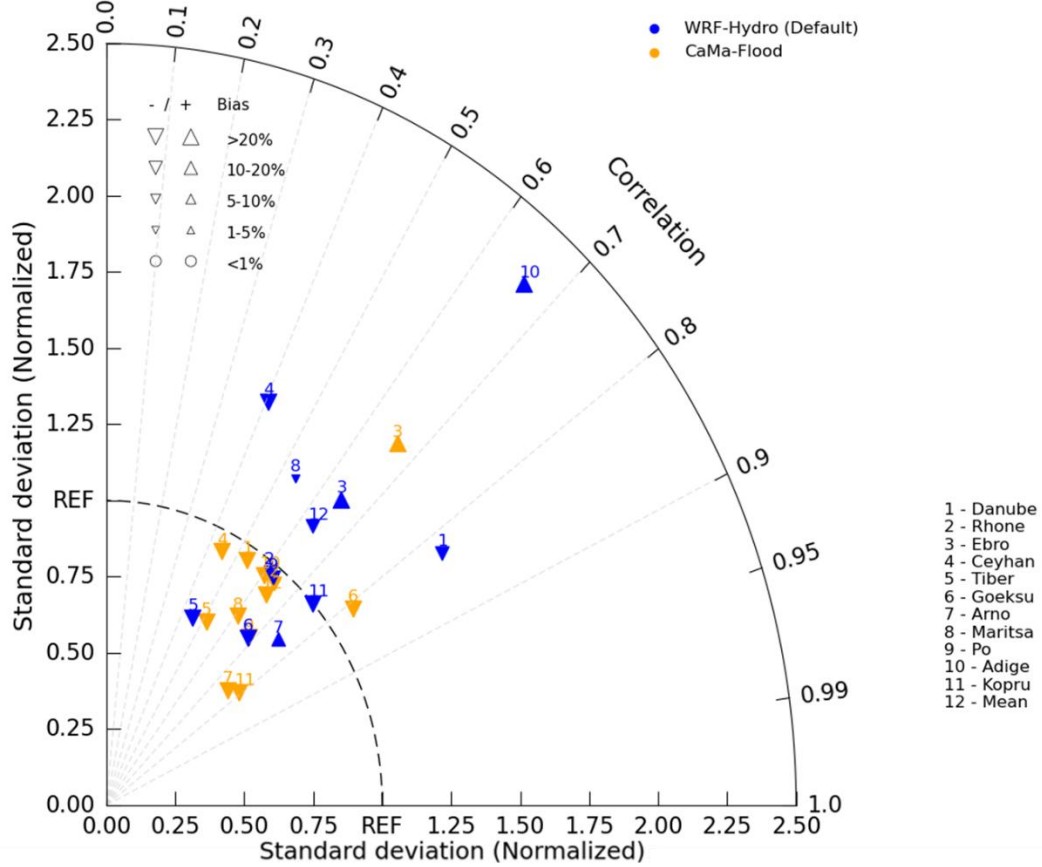

**Figure 2: Taylor diagram showing the performance of CaMa-Flood (orange) and default WRF-Hydro (Default, blue) models in different river basins numerated from 1 to 11. Number 12 refers to the mean value. The size and the orientation of the triangles**
**represent the magnitude and under- or overestimation of the percent bias.**

Spearman's rho, which evaluates flow timing, reveals mixed performance between CaMa-Flood and WRF-Hydro. Across all basins, the overall average Spearman's rho is nearly identical for both models, with CaMa-Flood at 0.64 and WRF-Hydro at 0.63 (median 0.64 and 0.65 respectively), indicating comparable performance in capturing flow timing. In several individual basins, the two models exhibit similar performance, such as the Ebro (CaMa-Flood: 0.66, WRF-Hydro: 0.65), Arno (0.76 vs.

0.75), and Rhone (0.64 vs. 0.61). However, in some basins, one model clearly outperforms the other. For instance, WRF-Hydro achieves noticeably better timing in the Danube (0.83 vs. 0.54), while CaMa-Flood shows superior performance in the Goeksu basin (0.81 vs. 0.68). Both models generally meet the threshold for acceptable performance (Spearman's rho > 0.5), indicating reasonable flow timing for most basins. However, significant differences in performance in basins like the Danube and Goeksu suggest that model-specific parameterizations and routing schemes might influence timing accuracy.

Across all basins, the overall average relative standard deviation is 0.90 for CaMa-Flood and 1.18 for WRF-Hydro, while the median values are 0.93 and 1.00 respectively. This indicates that CaMa-Flood generally underestimates variability, while WRF-Hydro shows a more centred distribution, but leans toward slightly higher variability. For CaMa-Flood, only five



basins fall within the acceptable range (i.e. 0.8-1.2), namely the Danube, Rhone, Ceyhan, Adige, and Goeksu, while the majority are below 1. Notable examples of this underestimation include the Arno (0.58), Maritsa (0.78), and Tiber (0.70)

basins, where variability is significantly lower than observed. In contrast, WRF-Hydro exhibits a more equilibrated performance, with six basins showing standard deviations higher than 1 and five basins lower than 1. However, only four basins—Rhone, Po, Arno, and Kopru—fall within the acceptable range. Despite its relatively balanced distribution, WRF-Hydro often displays excessive variability in basins where standard deviation values exceed 1. For instance, in the Danube (1.47), Adige (2.28), and Maritsa (1.27) basins, the variability is markedly overestimated, with the Adige basin standing out

as an extreme case of overestimation. Overall, CaMa-Flood demonstrates a consistent tendency to underestimate variability, with relatively few basins achieving balanced performance. WRF-Hydro, on the other hand, shows a broader variability, with a more equilibrated performance across basins but also a noticeable tendency toward excessive variability in some cases, occasionally to extreme levels. This distinction highlights fundamental differences in how the two models handle flow variability across the studied basins.

Across all basins, the overall average percent bias is -27.8% for CaMa-Flood and -12.1% for WRF-Hydro, while the median values are -26.6% and -17.1%, respectively. This result indicates that both models generally tend to underestimate flows, with CaMa-Flood exhibiting a more pronounced underestimation compared to WRF-Hydro. For CaMa-Flood, the percent bias exceeds -20% in most basins, such as the Ceyhan (-48.26%) and Kopru (-69.00%), underscoring a systematic tendency to underestimate flow. In contrast, WRF-Hydro demonstrates limited underestimation in certain basins, such as the Rhone (-

8.47%) and Maritsa (-3.98%), while overestimates flow in others, such as the Adige (64.63%) and Arno (19.42%). These findings highlight that while WRF-Hydro generally achieves lower bias, it may introduce overestimation in certain cases.

The Kling-Gupta Efficiency (KGE) offers a more complete performance metric than traditional benchmarks by combining these three aspects (i.e., correlation, bias and relative variability). A KGE value greater than −0.41 indicates better performance than the average streamflow benchmark, which represents the simplest model where the simulated value is

always the mean flow (Knoben et al., 2019). Furthermore, KGE values can be classified to indicate the quality of performance: values $\leq$ −0.41 are unacceptable, −0.41 < KGE $\leq$ 0.00 are very poor, 0.00 < KGE $\leq$ 0.30 are poor, 0.30 < KGE $\leq$ 0.65 are intermediate, and 0.65 < KGE $\leq$ 1.00 are good (adapted from Sanchez Lozano et al. (2025)).

The overall average KGE values for the two models across all basins are 0.37 for CaMa-Flood and 0.32 for WRF-Hydro, with median values of 0.43 and 0.45, respectively. This result indicates that both models generally achieve intermediate

performance, but with significant variability across basins (Fig. 3).





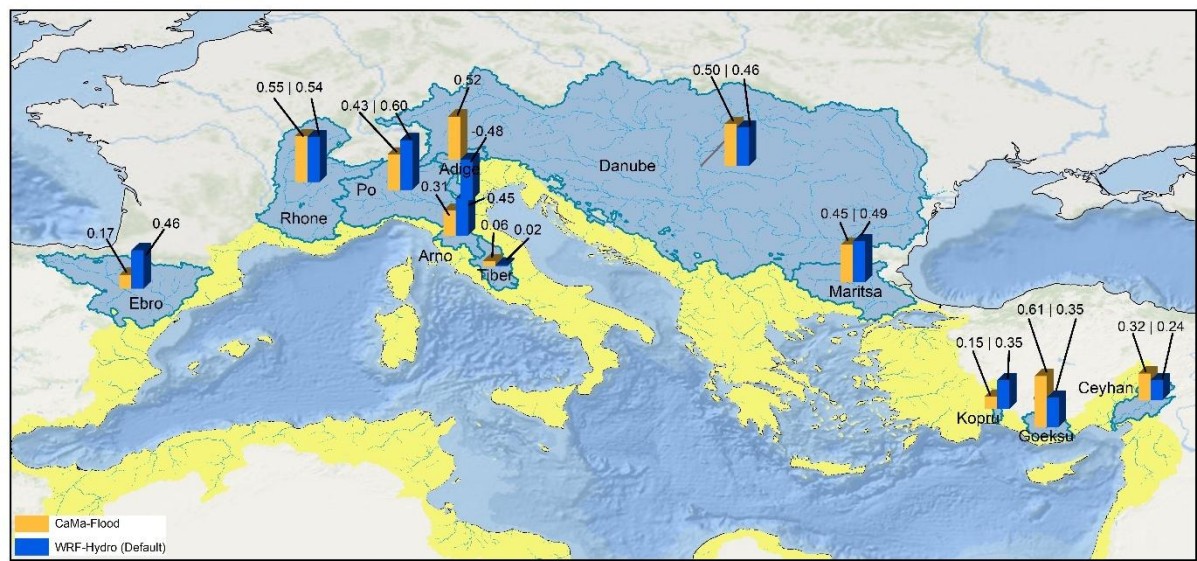

**Figure 3: KGE values of each model experiment (CaMa-Flood vs. WRF-Hydro (Default)) across the studied basins.**

For the CaMa-Flood model, the highest KGE value is observed in the Goeksu basin, falling within the "intermediate" performance category as the majority of the remaining basins (i.e., Rhone, Danube, and Adige). However, some basins, such

as the Tiber, Kopru, and Ebro, exhibit "poor" values, indicating limited predictive accuracy in these regions. WRF-Hydro shows a slightly more varied performance. Its highest values are observed in the Po basin and Rhone basin, both of which fall within the "intermediate" category as most of the other rivers. However, the Ceyhan and Tiber basins fall into the lower "poor" category ($0.00 < \text{KGE} \leq 0.30$), reflecting weaker predictive capability in these regions. Notably, the model significantly underperforms in the Adige basin, where its performance is categorized as "unacceptable," highlighting a key

area for improvement. When comparing the two models, WRF-Hydro performs better than CaMa-Flood in basins such as the Po, Arno, and Ebro, while CaMa-Flood outperforms WRF-Hydro in Goeksu, Ceyhan, and Adige basins. In some basins, i.e. the Danube, Maritsa, and Rhone, the two models exhibit comparable performance. Overall, the analysis highlights that neither of the models achieves intermediate or good performance across all basins, but each model shows strengths and weaknesses depending on the specific region.

In addition to the Spearman's rho, the time lag analysis offers valuable insights into the models' ability to capture the timing of peak flow events. The overall average time lag across all basins is 8.5 days for CaMa-Flood and 4.8 days for WRF-Hydro, with median values of 6 days for both models. These statistics highlight that WRF-Hydro generally achieves better timing accuracy, though both models exhibit variability across basins (Table 2).

**Table 2: Summary of time lag values for each model experiment (CaMa-Flood vs. WRF-Hydro (Default)) and each basin.**

| Basin | CaMa-Flood | WRF-Hydro (Default parameters) |
|---|---|---|
| Danube | 43 | 0 |



| Rhone | 5 | 6 |
|---|---|---|
| Po | 6 | 2 |
| Ceyhan | 3 | 3 |
| Adige | 4 | 3 |
| Tiber | 6 | 9 |
| Maritsa | 8 | 7 |
| Goeksu | 1 | 6 |
| Arno | 6 | 6 |
| Kopru | 1 | 4 |
| Ebro | 10 | 7 |


For the Danube basin, WRF-Hydro demonstrates exceptional accuracy with no time lag (0 days), while CaMa-Flood exhibits a remarkable lag of 43 days, highlighting a major discrepancy in flow timing for this basin. This result aligns with findings from Hamitouche et al. (2025). By analysing the ENEA-REG and WRF-Hydro simulated runoffs (Fig. S4 in the supplement), they observed a close alignment in timing, indicating that the coupling of the land surface model and routing in

WRF-Hydro does not significantly affect runoff timing. This suggests that the substantial lag in CaMa-Flood is primarily due to its routing limitations. In contrast, for most other basins, the two models show relatively similar performance, with time lags generally within a few days of each other. For instance, in the Rhone basin, both models exhibit minimal differences, with CaMa-Flood showing a lag of 5 days and WRF-Hydro 6 days. Similarly, in the Po, Ceyhan, Adige, and Arno basins, both models perform comparably, with time lags ranging from 2 to 6 days compared to observations. These results indicate

that in these regions, the models are generally aligned in predicting peak flow timing. However, notable discrepancies arise in specific basins. In the Goeksu basin, CaMa-Flood achieves near-perfect timing with a lag of just 1 day, whereas WRF-Hydro lags by 6 days. Similarly, in the Tiber basin, WRF-Hydro exhibits a higher lag of 9 days compared to 6 days for CaMa-Flood, suggesting slightly better timing by CaMa-Flood in this case. In summary, while both models show comparable performance in many basins, WRF-Hydro consistently demonstrates superior timing in basins like the Danube,

whereas CaMa-Flood achieves better timing in basins such as Goeksu. The large lag observed in the Danube for CaMa-Flood underscores a critical limitation of this model in capturing flow timing in specific regions.

Regarding low-flow and high-flow biases, these hydrological signatures evaluate performance using two segments of the flow duration curve (FDC) (Smakhtin, 2001; Vogel and Fennessey, 1994), as defined by Yilmaz et al. (2008). The high-flow segment (0–0.02 exceedance probabilities) represents watershed response to large precipitation events (Fig. S5 in the

supplement) and is used to calculate high-flow bias (%BiasFHV, Eq. (6)). The low-flow segment (0.7–1.0 exceedance





probabilities) reflects long-term flow sustainability (Fig. S6 in the supplement) and is used to calculate low-flow bias (%BiasFLV, Eq. (5)).

The evaluation of high-flow and low-flow biases reveals significant variability in the performance of CaMa-Flood and WRF-Hydro across basins (Fig. 4). For high-flow bias, the overall average bias across all basins is -27.8% for CaMa-Flood and -
2.1% for WRF-Hydro, with median values of -34.4% and -20.3%, respectively. This result indicates that CaMa-Flood generally underestimates high flows more frequently, while WRF-Hydro exhibits less pronounced biases but occasionally overestimates peak flows. For instance, WRF-Hydro shows large positive high-flow biases in basins like the Adige (89.25%) and Danube (34.83%). Conversely, CaMa-Flood tends to underestimate high flows substantially in basins such as the Kopru (-52.05%), Po (-44.95%), and Tiber (-42.89%). Notably, the Ebro basin stands out as the only instance where both models
demonstrate positive high-flow biases, especially for CaMa-Flood (33.07% vs. 12.43% in WRF-Hydro).

Regarding low-flow biases, the overall average bias across all basins is -16.3% for CaMa-Flood and -18.0% for WRF-Hydro, with median values of -17.3% and -16.7%, respectively. Both models predominantly show negative low-flow biases, indicating an underestimation of base flows. However, the extent of these biases varies. For example, in the Tiber and Goeksu basins, WRF-Hydro exhibits significantly larger negative biases (-41.99% and -35.06%, respectively) compared to
CaMa-Flood (-23.21% and -24.02%). On the other hand, WRF-Hydro shows positive low-flow biases in the Adige (9.74%) and Arno (14.44%) basins, suggesting occasional overestimations, whereas CaMa-Flood consistently maintains negative biases in these basins, with values of -9.70% for the Adige and -2.63% for the Arno.

Overall, CaMa-Flood demonstrates a more consistent tendency to underestimate both high and low flows across most basins, reflecting a conservative approach in capturing variability. In contrast, WRF-Hydro shows more mixed behaviour, with both
significant overestimations (e.g., Adige and Danube for high flows) and underestimations (e.g., Tiber and Goeksu for low flows), highlighting its variable performance across different flow regimes and basins.



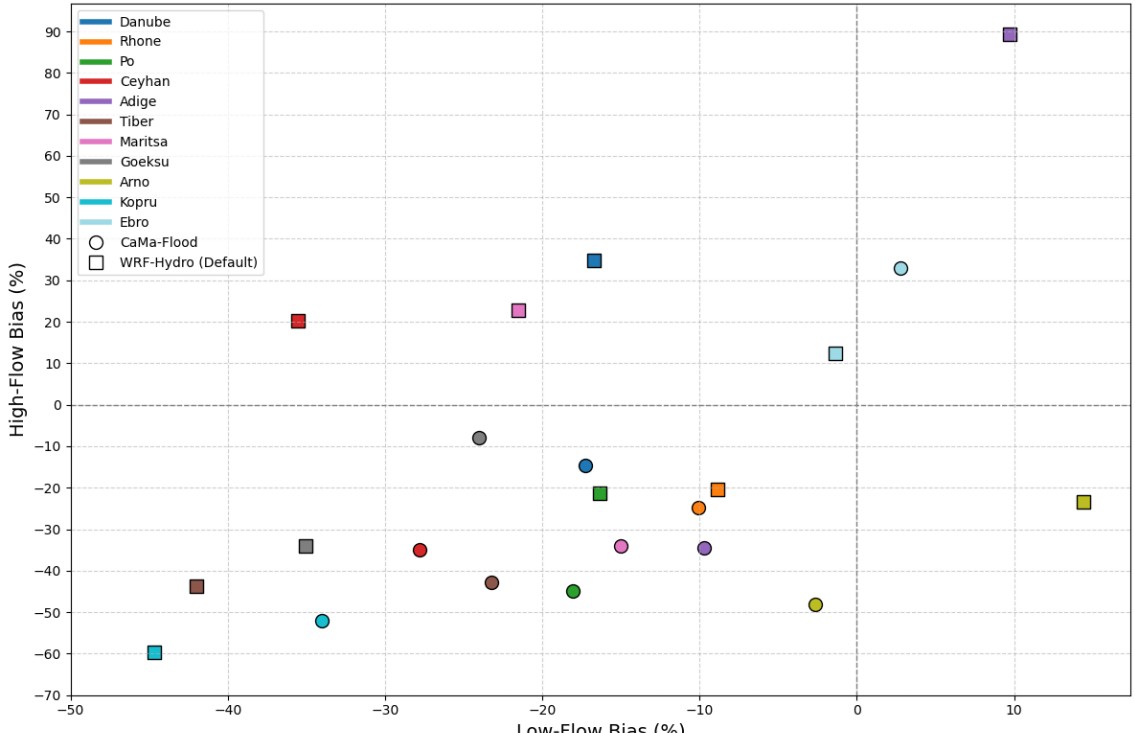

**Figure 4: Scatterplot of low-flow bias vs. high-flow bias in each model experiment (CaMa-Flood vs. WRF-Hydro) and river basin.**

To explore whether the basin size influences the model performance, we analysed the Pearson's correlation between key
performance metrics and the size of the analysed Mediterranean river basins. These basins represent a wide range of
upstream areas, from smaller basins like Kopru to larger ones such as the Rhone. The Danube basin was excluded from this
analysis due to its significantly larger size compared to the others, which could substantially alter the correlation results. The
results for Spearman's rho indicate that the flow timing of both CaMa-Flood and WRF-Hydro models shows a weak
relationship with basin size, with both models having a correlation value of -0.13. The flow variability, as well as the time
lag, increases with increasing basin size only for CaMa-Flood (correlation of 0.48 and 0.63 respectively). For percent bias,
CaMa-Flood is less sensitive to basin size compared to WRF-Hydro (-032 vs. -0.67), especially in terms of high-flow bias (-
0.18 for CaMa-Flood, -0.58 for WRF-Hydro) compared to low-flow bias (-0.46 for CaMa-Flood, -0.66 for WRF-Hydro).
Although these correlations are not statistically valid due to the limited number of basins analysed, they are still useful for
gaining insight into the sensitivity of each model to basin size.

## 3.2 Impact of calibration on WRF-Hydro model performance

The calibration of WRF-Hydro, aimed to optimize the Kling-Gupta Efficiency (KGE), leads to an improvement of the model
performances in most of the basins and metrics. In average, the calibration improved the KGE from 0.32 to 0.48, with the
median increasing from 0.45 to 0.60. The quality of performance pass from intermediate to good in several basins such as



the Arno (from 0.45 to 0.66), the Danube (from 0.46 to 0.66), the Rhone (from 0.54 to 0.65) and the Ebro (from 0.46 to
0.65). After calibration, Goeksu and Tiber rivers show intermediate performance, while for the Po, Kopru, and Maritsa
basins the KGE remained unchanged (Fig. 5).

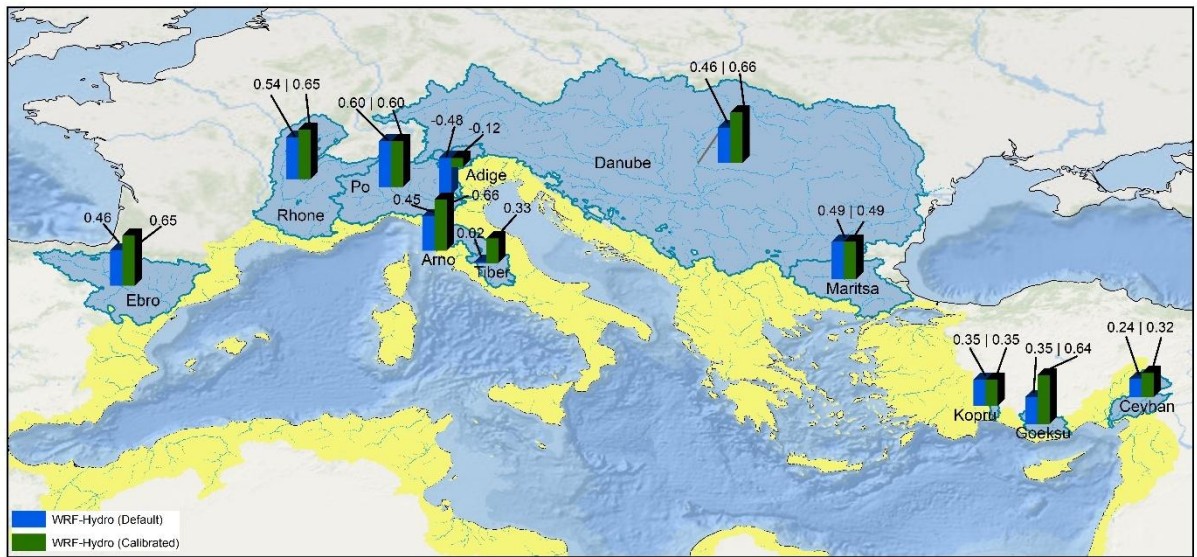

**Figure 5: Comparison of KGE values between calibrated and non-calibrated WRF-Hydro model across the selected basins.**

The components of the KGE offer further insight into these outcomes. The temporal correlation showed relative stability
across most basins, with modest improvements in the Goeksu (from 0.68 to 0.76), Rhone (0.61 to 0.67) and Arno (from 0.75
to 0.80), while noticeable declines were found in the Ceyhan (from 0.41 to 0.28). These results indicate that the improved
temporal alignment between modelled and observed discharges is not systematic for all basins. In terms of bias, calibration
yielded mixed results: the bias reduced substantially for the Goeksu (-33%) and the Rhone (-8.2%) basins, while worsen in
the Danube (+4.6%) and the Ceyhan (+8.2%). These changes illustrate that while calibration reduced bias in many cases, it
occasionally introduced trade-offs. The standard deviation also exhibited changes with an average decreased from 1.18 to
1.08, suggesting improved consistency between simulated and observed flow variability. Relevant improvements were seen
in basins such as the Danube and Ceyhan, where standard deviation dropped from 1.47 to 0.83 and from 1.45 to 0.99
respectively, and in basins like the Arno, where standard deviation increased from 0.83 to 1.04 (Fig. 6).





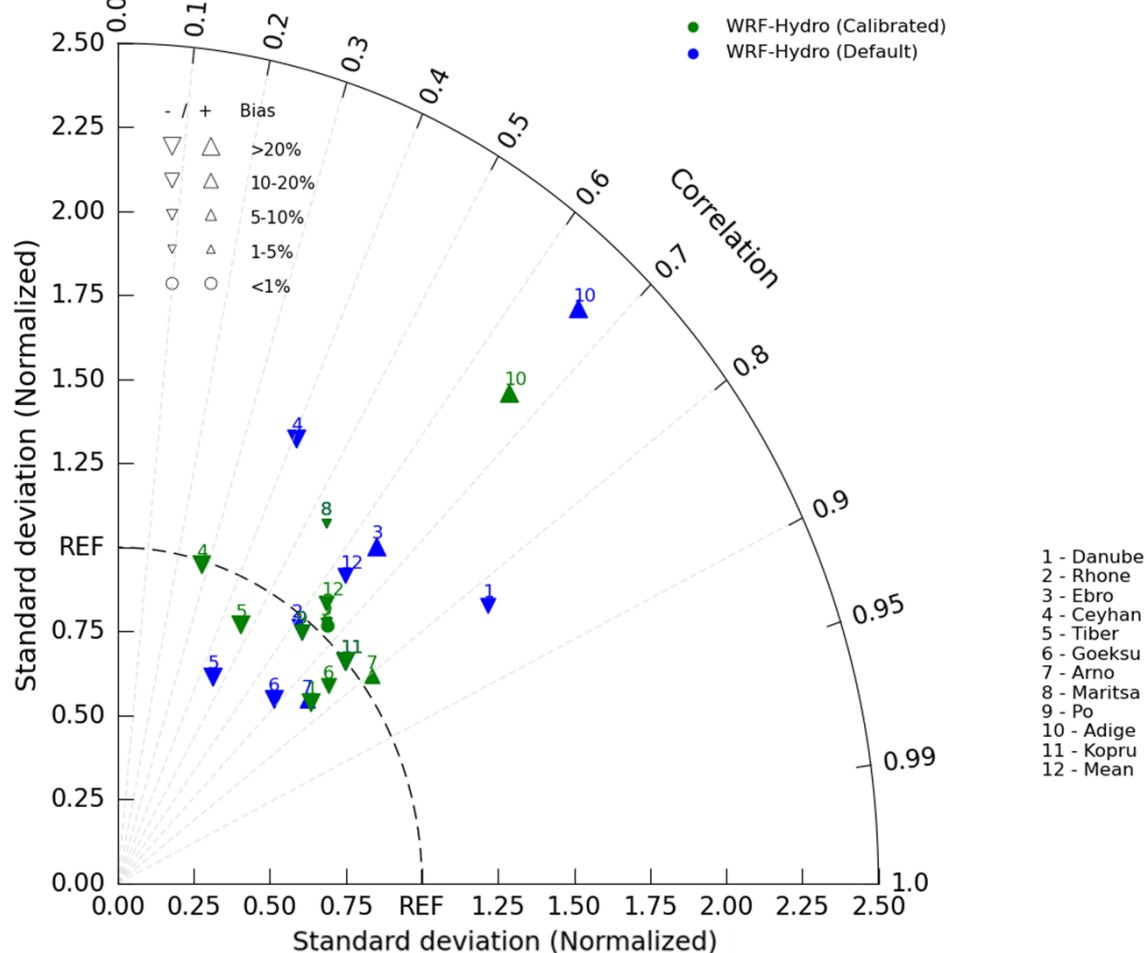

**Figure 6: Taylor diagram showing the performance of WRF-Hydro (Default, blue) and WRF-Hydro (Calibrated, green) models in different river basins numerated from 1 to 11. Number 12 refers to the mean value. The size and the orientation of the triangles represent the magnitude and under- or overestimation of the percent bias.**

In addition to these core metrics, calibration also influenced lag time (Table 3), low-flow bias, and high-flow bias (Fig. 7). Lag time improved noticeably with the average decreasing from 4.8 to 2 days. The Rhone, Ceyhan, and Arno showed substantial reductions, achieving zero lag time after calibration, while slight increases were observed in basins like the Danube, where lag time rose from 0 to 3 days. Low-flow bias exhibited minor overall improvement, shifting in average from -18.0% to -17.3%, with notable reductions in the Rhone and Goeksu basins, but with a worsening in Ceyhan. High-flow bias, on the other hand, displayed mixed results: a reduction of the extreme positive biases in the Danube, (from 34.8% to -11.9%) and the Ceyhan (from 20.1% to -6.8%) together with a slight worsening in the Adige basin (from 89.3% to 90.9%).

**Table 3: Summary of time lag values for calibrated and non-calibrated WRF-Hydro model in each basin.**

| Basin | WRF-Hydro (Default parameters) | WRF-Hydro (Calibrated) |
|-------|-------------------------------|------------------------|





| Danube | 0 | 3 |
|--------|---|---|
| Rhone | 6 | 1 |
| Po | 2 | |
| Ceyhan | 3 | 0 |
| Adige | 3 | 0 |
| Tiber | 9 | 0 |
| Maritsa | 7 | |
| Goeksu | 6 | 3 |
| Arno | 6 | 0 |
| Kopru | 4 | |
| Ebro | 7 | 0 |

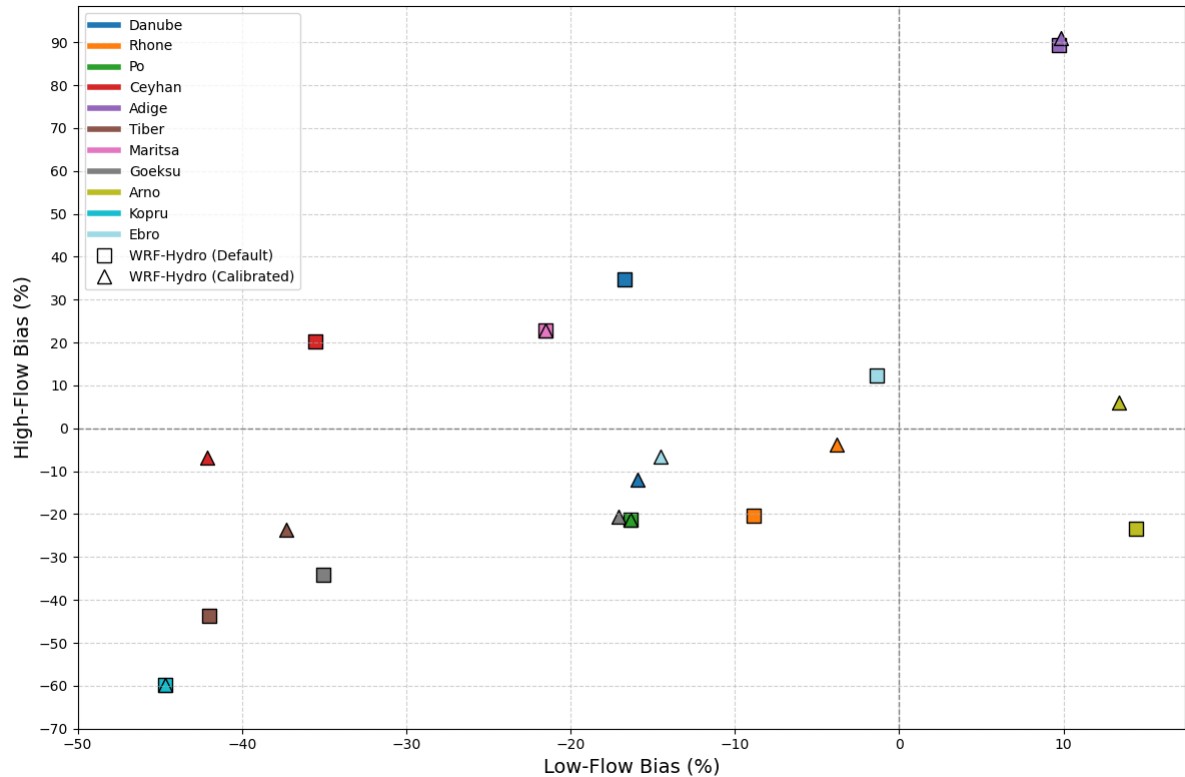

**Figure 7: Scatterplot of low-flow bias vs. high-flow bias for calibrated and non-calibrated WRF-Hydro model in each river basin.**

The unchanged KGE values for the Po, Kopru, and Maritsa basins are particularly noteworthy. This finding suggests that the

default parameters for these basins were already optimal, and further calibration did not yield any improvements. This





outcome highlights the robustness of the default parameter configuration in specific contexts and emphasizes the need for calibration strategies that are sensitive to basin-specific hydrological characteristics. Overall, the calibration of WRF-Hydro enhanced discharge simulations for many Mediterranean basins, as reflected in the improvements in KGE and reductions in lag time. However, the trade-offs between correlation, bias, and standard deviation, along with the basin-specific variability,

highlight areas where the calibration strategy could be refined to better capture the unique dynamics of each basin.

For the Adige basin, the poor performance, both with default and calibrated WRF-Hydro, observed particularly for the strong overestimation of high-flow events, the overall bias, and the variability, resulting in a very poor KGE, can be explained by the substantial corrections made to the drainage network. To disconnect the Adige basin from the adjacent Po River basin— necessary due to the coarse spatial resolution of the model—elevation reconditioning was applied. However, these

corrections altered the topography slope, a critical factor in discharge calculations. The changes to the slope resulted in overly fast flows, which in turn caused exaggerated high-flow peaks and impacted the model's ability to simulate discharge accurately for the Adige basin. This finding underscores the sensitivity of WRF-Hydro, as any other hydrodynamic or hydrological model, to topography and the need for careful elevation reconditioning, particularly at coarse spatial resolutions. While such corrections are essential to accurately delineate basin boundaries and drainage networks, they must

be implemented cautiously to avoid introducing errors that affect hydrological dynamics.

Excluding the Adige basin from the analysis demonstrated notable improvements in the overall performance of the WRF-Hydro model outperforming CaMa-Flood (Fig. 8). For instance, the calibrated WRF-Hydro configuration showed an increase in KGE from 0.48 (including Adige) to 0.53 (excluding Adige), compared to the default settings (0.32 and 0.4 respectively) and CaMa-Flood (0.37). Similarly, the percent bias improved from 23.9% to 21.5%, and the relative standard

deviation became closer to 1, indicating better approximation of flow variability. These improvements underscore the sensitivity of WRF-Hydro's performance to accurate input parameters and the profound impact of specific basin corrections on overall model results.



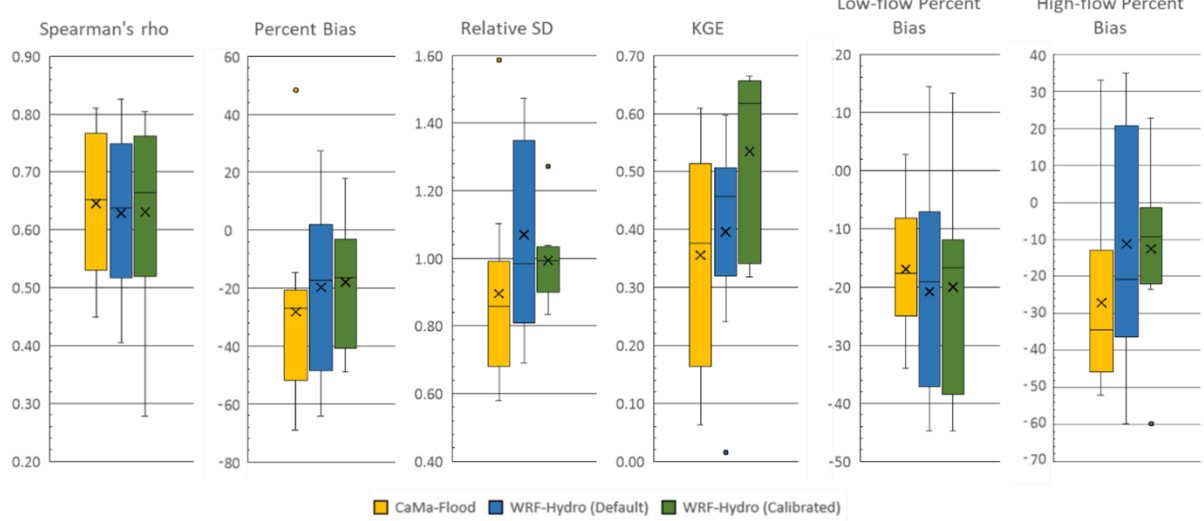

**Figure 8: Performance metrics for each model experiment (with Adige basin excluded). The X represents the mean value.**

It is worth noting that while some metrics, such as high-flow bias, showed a decrease in performance, this is primarily because the Adige basin's overestimation of high flows compensated for the negative biases in other basins. Nevertheless, the improvements in the key metrics—KGE, Percent Bias, and relative standard deviation—demonstrate the importance of addressing basin-specific challenges to enhance model reliability.

Details on the calibrated parameters, their influence on key hydrological processes, and observed trends across basins are
provided in the Supplement (Sect. S1 and Table S4).

To analyse the combined effect of the calibrated parameters, we examined the simulated runoff, for possible comparison, as it serves as the primary input for CaMa-Flood. Table S5 in the supplement provides an overview of the average daily total runoff (mm/day) and the ratio of subsurface to total runoff simulated by ENEA-REG model and WRF-Hydro, using both default and calibrated parameter values.

Before delving into the impact of calibration, we first assessed the differences between runoff simulated by ENEA-REG and WRF-Hydro. This comparison is essential, as it contributes to understand the observed performance differences between the CaMa-Flood and WRF-Hydro models, alongside the distinct routing configurations in each model. Our results indicate that WRF-Hydro predominantly generate higher runoff compared to ENEA-REG, with percentage increases ranging from 11.1% to 49.8%, except for the Göksu basin, where WRF-Hydro shows lower runoff by 3.6%. Similarly, the subsurface-to-total
runoff ratio simulated by WRF-Hydro exceeds that of ENEA-REG across all basins, with percentage increases ranging from 4.1% to 119.9%. These findings underscore the sensitivity of runoff to the dynamic interactions between land surface models, routing processes, and the partitioning of surface and subsurface flows.

Turning to the influence of calibration, and excluding the Po, Maritsa, and Kopru basins—where default and calibrated parameters coincide—the calibration process led to a remarkable reduction in total runoff for most basins. This decrease,
brought runoff values closer to, or even below, those simulated by ENEA-REG with and average reduction between 20.5%



and 48.1%. However, in the Tiber and Göksu basins, the calibrated parameters produced a marked increase in total runoff. These variations can largely be attributed to changes in parameters that control transpiration and soil evaporation, as the slope parameter consistently decreased, limiting deep drainage.

For the subsurface-to-total runoff ratio, calibration effects varied. In the Göksu basin, there was a slight increase of 3.8%,
while for other basins the ratio decreased, either marginally (e.g., Danube at -1.5%, Adige at -3.4%, and Ceyhan at -4.9%) or significantly (e.g., Rhone at -26.2% and Ebro at -28.7%). These changes stemmed from the combined influence of parameters governing infiltration and runoff partitioning. Together, the results highlight the critical role of parameter calibration in modulating both total runoff and its partitioning into surface and subsurface components, thereby influencing hydrological model performance.

Post-calibration improvement was actually expected, particularly because locally calibrated scores help compensate for errors in catchment boundaries (Kauffeldt et al., 2013; Lehner, 2012), primarily due to the coarse resolution of elevation data, as exemplified by the Adige basin, and for errors in meteorological forcing, particularly in precipitation data (Beck et al., 2017, 2019), as shown in Sect. S2 and Fig. S7 in the supplement.

**4 Conclusions**

This study assessed whether WRF-Hydro and CaMa-Flood can serve as effective alternatives for improving hydrological simulations within Euro-Mediterranean regional coupled models and evaluated the extent to which calibration can enhance WRF-Hydro's performance.

In their default configurations, both models demonstrated potential to improve hydrological simulations, though their effectiveness varied across basins. CaMa-Flood excelled in computational efficiency and exhibited reasonable timing
accuracy in certain basins, such as Goeksu. However, it consistently underestimated flow variability and high-flow extremes and showed notable limitations in large basins, such as the Danube, where significant timing delays occurred. These findings indicate that while CaMa-Flood provides a computationally efficient option for regional coupled models, but enhancements in its routing scheme are necessary to better capture flow variability and peak events.

WRF-Hydro, in its default configuration, achieved comparable or superior performance to CaMa-Flood in many basins,
particularly in capturing flow timing and reducing biases. For example, it performed better than CaMa-Flood in the Danube, where it achieved no timing lag compared to CaMa-Flood's 43-day delay. These findings demonstrate that both models can serve as viable alternatives for hydrological simulations, but their utility depends on basin-specific characteristics, the trade-offs between computational cost and simulation accuracy, and the distinct advantages offered by each model. For studies focusing on climate impacts on flooding and flooded areas, CaMa-Flood is particularly well-suited as it allows for the
simulation of floodplain inundation dynamics. Although it exhibits higher average bias, this bias remains within a reasonable range when applied to large, managed Mediterranean river basins where observed streamflow often deviates from natural flow conditions. Conversely, WRF-Hydro, with its full coupling between soil hydrology and the atmosphere, is ideal for



studies investigating precipitation-soil moisture feedback mechanisms. These complementary strengths emphasize the importance of aligning model selection with the objectives of the study.

Calibration significantly improved the performance of WRF-Hydro across most basins. Key metrics such as Kling-Gupta Efficiency (KGE) and timing lag demonstrated substantial improvements, with average KGE increasing from 0.32 to 0.48 and timing lags reducing from 4.8 days to 2 days. Notable improvements were observed in basins like the Rhone and Arno, where calibration reduced bias and improved flow variability alignment with observations. However, challenges remained in basins like the Adige, where coarse-resolution topographical corrections introduced errors that limited the effectiveness of

calibration. These findings emphasize the need for careful preparation of model inputs, particularly at coarser resolutions, to ensure reliable simulations.

The calibrated parameters derived in this study hold promise for broader application. They can be extrapolated to other homogeneous basins with similar hydrological and climatic characteristics or used as a first-stage calibration for higher-resolution simulations. Although regionalization was not performed due to limited validation data, these results provide a

valuable foundation for improving discharge simulations in other regions.

It is worth noting that this calibration is primarily intended for offline atmosphere-hydrological simulations to close the water cycle at the land-ocean interface. For studies aimed at assessing the feedback and impact of fully coupled atmosphere-hydrological models, a tailored calibration approach based on observed meteorological inputs should be considered.

In conclusion, both WRF-Hydro and CaMa-Flood, in their default configurations, can serve as viable alternatives for

improving hydrological simulations within Euro-Mediterranean regional coupled models. WRF-Hydro generally offers greater potential, especially after calibration, which significantly enhances its performance. By leveraging the distinct advantages of each model, such as CaMa-Flood's ability to simulate floodplain inundation and its reasonable bias for disturbed basins, and WRF-Hydro's coupling between soil hydrology and the atmosphere, a basin-specific approach can improve hydrological predictions and projections.

**Code and data availability**

GRDC discharge observations can be obtained from https://grdc.bafg.de/data/data_portal/ (GRDC, 2024). ERA5 data, provided by the European Centre for Medium-Range Weather Forecast (ECMWF), can be freely downloaded from the Copernicus Data Store (https://doi.org/10.24381/cds.adbb2d47 (Hersbach et al., 2020)). ENEA-REG, WRF-Hydro and CaMa-Flood models can be downloaded from https://github.com/uturuncoglu/RegESM/releases/tag/1.2.0_enea (Anav et al.,

2021),        https://github.com/NCAR/wrf_hydro_nwm_public/releases/tag/v5.2.0        (Gochis        et        al.,        2021)        and https://github.com/global-hydrodynamics/CaMa-Flood_v4 (Yamazaki et al., 2011, 2013), respectively. The WRF-Hydro calibration package, is available at: https://github.com/NCAR/PyWrfHydroCalib, or upon request from the NCAR WRF-Hydro team (wrfhydro@ucar.edu). Model output data are available from the corresponding author upon request due to their

large volume. However, a step-by-step instruction file, along with all associated materials required to reproduce the results of

this study, is publicly available on Zenodo at https://doi.org/10.5281/zenodo.16333943 (Hamitouche et al., 2025).

**Author contribution**

MH, GF, and AA: conceptualisation and methodology; MH: formal analysis, funding acquisition, investigation, visualisation and writing-original draft preparation; MH and AA: simulations; AR: helped with calibration setup; GF and AA: supervision; GF, AA, and AR: validation and writing-review & editing. All authors have read and agreed to the published

version of the paper.

**Competing interests**

The authors declare that they have no conflict of interest.

**Acknowledgements**

We would like to thank Stefan Hagemann for providing us with some discharge observation data and Day Yamazaki and

Kevin Samspson for their valuable help in setting up the CaMa-Flood and WRF-Hydro models.

**Financial support**

This paper and related research have been conducted during and with the support of the Italian PhD course in Sustainable Development and Climate change (link: www.phd-sdc.it) at the University School for Advanced Studies IUSS and developed within the framework of the project "Dipartimento di Eccellenza 2023-2027"., with the financial support from

the ICSC Italian Research Center on High-Performance Computing, Big Data and Quantum Computing and received funding from the European Union Next-GenerationEU (National Recovery and Resilience Plan-NRRP, Mission 4, Component 2, Investment 1.4-D.D: 3138 16/12/2021, CN00000013).

This research has also been supported by the CIHEAM Prize for the Best MSc Thesis 2022.

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
