# Peer review of "Regional-scale Hydrologic Model Comparison Including Calibration for Improved River Discharge Simulations into the Mediterranean Sea"

_EGUsphere, 2025_

## Author Comment (AC2)

The authors offer a regionally-focused comparison of two hydrological routing models, CaMa-Flood and WRF-Hydro, using default parameters, as well as a calibrated WRF-Hydro. The paper is a nice contribution to the Mediterranean region hydro modeling scientific literature. I suggest Major Revision, with three Major comments and several Minor comments.

Major comments:

1) **Compare results with a baseline**. To show improvement in river discharge simulations, it would be helpful to compare the results with an existing baseline/benchmark. For instance, in the introduction, part of the motivation was that the HD model was underestimating discharge in the Med. Similarly, CaMa and WRF-Hydro also tended to underestimate discharge. Is there any way to see if either model actually offered an improvement? Even if it is qualitative? If not, this point should still be mentioned as a limitation of the work.

> - *We thank the reviewer for this valuable suggestion. To address it, we added a dedicated discussion of the HD model (which is the default river-routing component in the current ENEA-REG configuration) as a baseline benchmark in Section 3 ("Results and Discussion"). Specifically, before comparing the performances of CaMa-Flood and WRF-Hydro, we now present both a qualitative and quantitative evaluation of HD simulations against observations. This includes discharge time series (as Fig. S4 in the supplement) to visually contrast HD results with observations, CaMa-Flood, and WRF-Hydro, and statistical metrics averaged across all Mediterranean basins, with basin-scale details provided in the Supplement (as Tables S4 and S5).*

*3 Results and discussion:*

*"Before comparing the performances of CaMa-Flood and WRF-Hydro, we first present their simulated discharge time series, alongside observations and the discharge simulated by the HD model at both 0.5-degree and 5-minute spatial resolutions (Fig. S4). The higher-resolution HD configuration is intended to improve the representation of the drainage network and catchment area, thereby enhancing discharge estimates. However, both HD versions show a clear underestimation of freshwater inflow to the Mediterranean Sea and fail to reproduce the observed temporal patterns.*

*Quantitatively, the HD model exhibits very poor performance across Mediterranean basins. At 0.5-degree, the overall average KGE is –0.13, with a mean bias of –37.4%, a Spearman's correlation of 0.36, and a relative SD of only 0.23. At 5-minute, the performance improves slightly (KGE = 0.00; %Bias = –16.6%; $r_s$ = 0.48), but variability remains strongly damped (rSD = 0.25). The most critical shortcoming is the systematic underestimation of extremes: high-flow biases reach –70.3% (0.5-degree) and –65.7% (5-minute), while low-flow biases are also negative (–25.3% and –9.2%, respectively). Model performances at the basin scale for both HD versions (0.5-degree and 5-minute) are provided in Tables S5 and S6 of the Supplement, respectively, offering basin-specific details beyond the aggregated statistics presented here.*

*These results suggest that the underestimation of freshwater inflow by the HD model is not primarily due to the long-term mean bias—which is moderate in the 5-minute configuration—but rather to its inability to capture discharge variability and high-flow peaks.*

*In contrast, visually WRF-Hydro and CaMa-Flood reproduce observed temporal dynamics much more closely, including variability and extremes, and therefore clearly outperform the HD*

*model. Both models show strong potential for capturing observed patterns when using runoff from the Noah-MP land surface model—the default LSM in many Euro-Mediterranean regional coupled and Earth system models—which has already been validated against ERA5-Land runoff data (Hamitouche et al., 2025). This indicates that CaMa-Flood and WRF-Hydro are suitable alternatives to replace the HD model in this modelling framework, despite some visual biases in low- and high-flow reproduction. The following section provides a detailed evaluation of their performances."*

[Figure]

*Figure S4: Observed and simulated daily discharge for the Ebro, Rhone, Tiber, Goeksu and Po rivers for common 10 years from 1995 to 2004. Simulations include ENEA-REG–driven WRF-Hydro and CaMa-Flood, as well as the HD model at 0.5° and 5-minute spatial resolutions, evaluated near the corresponding gauge stations.*

[Figure]

*Figure S4 (continuity): Observed and simulated daily discharge for the Kopru, Maritsa, Ceyhan and Danube rivers for common 10 years from 1999 to 2008. Simulations include ENEA-REG–driven WRF-Hydro and CaMa-Flood, as well as the HD model at 0.5° and 5-minute spatial resolutions, evaluated near the corresponding gauge stations.*

[Figure]

*Figure S4 (continuity): Observed and simulated daily discharge for the Adige and Arno rivers for common 10 years from 2005 to 2014. Simulations include ENEA-REG–driven WRF-Hydro and CaMa-Flood, as well as the HD model at 0.5° and 5-minute spatial resolutions, evaluated near the corresponding gauge stations.*

*Table S5: Summary of the performances (KGE, bias, correlation, relative standard deviation, low-flow and high-flow biases) of HD model at 0.5-degree spatial resolution for each river basin, evaluated over the entire 1990-2014 period. No values are reported for the Kopru basin, as its small size relative to the coarse 0.5-degree resolution prevented extraction of a representative discharge time series.*

| Basin | KGE | Percent bias | Spearman's rho | Relative SD | High-flow Percent Pbias | Low-flow Percent Pbias |
|-------|-----|-------------|----------------|-------------|-------------------------|------------------------|
| Danube | 0.15 | -21.88 | 0.35 | 0.42 | -40.62 | -12.44 |
| Rhone | -0.05 | -20.92 | 0.31 | 0.28 | -61.07 | -18.73 |
| Po | -0.08 | -42.61 | 0.52 | 0.18 | -77.32 | -42.33 |
| Ceyhan | -0.36 | -75.77 | 0.30 | 0.11 | -88.31 | -63.25 |
| Adige | 0.11 | -14.57 | 0.47 | 0.33 | -57.16 | -1.36 |

| | | | | | |
|---|---|---|---|---|---|
| Tiber | -0.47 | -89.30 | 0.40 | 0.10 | -89.02 | -67.41 |
| Maritsa | -0.01 | 1.56 | 0.32 | 0.27 | -65.12 | 6.91 |
| Goeksu | -0.49 | -83.36 | 0.26 | 0.05 | -94.07 | -66.58 |
| Arno | -0.18 | -29.77 | 0.32 | 0.13 | -84.83 | 8.13 |
| Kopru | / | | | | | |
| Ebro | 0.10 | 2.86 | 0.33 | 0.45 | -45.08 | 4.42 |

*Table S6 Summary of the performances (KGE, bias, correlation, relative standard deviation, low-flow and high-flow biases) of HD model at 5 minutes spatial resolution for each river basin, evaluated over the entire 1990-2014 period.*

| Basin | KGE | Percent bias | Spearman's rho | Relative SD | High-flow Percent Pbias | Low-flow Percent Pbias |
|---|---|---|---|---|---|---|
| Danube | -0.02 | -19.02 | 0.21 | 0.30 | -45.14 | -8.82 |
| Rhone | -0.05 | -15.30 | 0.31 | 0.22 | -64.42 | -6.55 |
| Po | -0.01 | -29.79 | 0.53 | 0.21 | -74.04 | -21.67 |
| Ceyhan | 0.12 | -27.36 | 0.50 | 0.32 | -64.56 | -7.97 |
| Adige | 0.18 | 38.65 | 0.55 | 0.44 | -37.52 | 16.82 |
| Tiber | -0.07 | -52.18 | 0.60 | 0.18 | -78.90 | -59.56 |
| Maritsa | 0.31 | 6.41 | 0.64 | 0.40 | -50.96 | 7.33 |
| Goeksu | -0.05 | -36.57 | 0.58 | 0.17 | -80.35 | -6.63 |
| Arno | -0.13 | -28.15 | 0.34 | 0.13 | -84.43 | 9.83 |
| Kopru | -0.22 | -68.48 | 0.64 | 0.10 | -89.17 | -61.27 |
| Ebro | -0.05 | 49.63 | 0.42 | 0.33 | -53.01 | 37.40 |

2) **Integrate the calibration results**. As written and presented, the calibrated WRF-Hydro results seem like a separate add-on, rather than an integrated component. Some integration occurred starting on line 466, but this was hard to follow.

- *We appreciate the reviewer's observation. Our intention was to present the calibration of WRF-Hydro as a complementary analysis rather than as part of the main model comparison. This decision was motivated by several factors:*
  1. *Unlike WRF-Hydro, CaMa-Flood and Noah-MP do not currently support basin-specific or spatially variable parameter calibration, and Noah-MP (within ENEA-REG) is not two-way coupled with CaMa-Flood. To ensure fairness in the core comparison between the two routing models, we restricted the main evaluation to non-calibrated configurations.*

> 2. **WRF-Hydro calibration was facilitated thanks to the availability of the PyWRFHydroCalib package, whereas no equivalent tool exists for CaMa-Flood in our framework.**
> 3. **Calibration is computationally expensive, and we considered it more appropriate to present the results as an "add-on" analysis, illustrating the potential benefits of calibration where resources allow, rather than as a standard step in the intercomparison.**
>
> **For these reasons, we structured the calibration results as a separate subsection, highlighting their added value while keeping the baseline comparison consistent and unbiased. We have revised the section headings to clarify this rationale and improve the readability of the calibration analysis.**

3) **Improve clarity, organization, and flow.** The authors include a lot of good content, but sometimes it was hard to follow and assumed prior familiarity. Specific suggestions for this are offered in the Minor Comments listed line-by-line, below.

> - **We thank the reviewer for this observation. We carefully revised the manuscript by addressing the suggestions provided in the minor comments, which helped us improve the clarity, organization, and flow of the paper.**

Minor comments:

1) Line 25. Suggest: "Not only does it provide essential freshwater input…"

> - **We thank the reviewer for the suggestion. The sentence has been revised accordingly to: "Not only does it provide essential freshwater input…"**

2) Line 30: Suggest tightening up this paragraph to better link with previous paragraph, which was only about the importance of the regional discharges in the Med Sea. Suggest revising this first sentence (also not sure if the Nearing reference fits, since that ref is more about AI-forecasting). Maybe: "Timely and accurate river discharge estimates into the Mediterranean are critical for managing water resources and risks in the region." Then second sentence could merge with the 3rd to be, "In a broader context, understanding the interplay between hydrological processes and regional climate is essential, and underscores the need to study the coupling of …."

> - **We thank the reviewer for the helpful suggestion. The paragraph has been revised to improve clarity and linkage, and now reads:**
> **"Timely and accurate river discharge estimates into the Mediterranean are critical for managing water resources and related risks in the region (Cisterna-García et al., 2025). In a broader context, understanding the interplay between regional climate change and hydrological processes is essential, and underscores the need to study the coupling of Earth system components, including atmospheric, hydrological, and oceanic processes.".**

3) Line 56. Suggest replacing "Nowadays". Maybe something about Recent advances?

- *We thank the reviewer for the suggestion. The sentence has been revised to:*
  *"Recent advances have led to the development of several complex coupled models, aimed at achieving fully integrated hydrological predictions for the Mediterranean region."*

4) Line 60-64. This has important information but is hard to follow for someone not very familiar with these models; suggest revising for clarity. What is HD? How do the different horizonal resolutions factor in? What is HydroPy LSM?

- *We thank the reviewer for pointing this out. To improve clarity, we first explicitly define the models: the HD model refers to the Hydrological Discharge model (Hagemann and Dümenil, 1997), which is a river routing model, while HydroPy is a Land Surface Model (LSM) (Stacke and Hagemann, 2021) that simulates runoff.*
  *The revised text now reads:*
  *"This underestimation seems to be with the Hydrological Discharge (HD) model (Hagemann and Dümenil, 1997), a river routing model used to simulate the river discharge with different horizontal resolutions (e.g. 5 minutes in MESMAR and 0.5 degrees in ENEA-REG coupled models). The HD model uses a pre-parametrization based on a linear reservoir routing concept with pre-defined reservoir numbers and temporal constants tailored to runoff inputs from the HydroPy Land Surface Model (LSM) (Stacke and Hagemann, 2021), which neglects the energy budget and overestimates runoff."*
  *We also clarify later in Section 3 ("Results and discussion") that the higher-resolution HD configuration (5-minute) is intended to improve the representation of the drainage network and catchment area, thereby enhancing discharge estimates (see reply to major comment 1).*

5) Line 75. About this sentence -> "This highlights the importance of conducting detailed sensitivity analyses on standalone routing models, before adding them into coupled climate or Earth system models." <- I think this is too general, and what is the sensitivity analysis here? Seems more like a comparative evaluation. Also, can you link this to a specific regional model or modelling experiment, such as the Med-CORDEX experiment, rather than the general coupled ESMs? Or later when you say "Euro-Mediterranean regional coupled models"?

- *We thank the reviewer for pointing this out. We agree that the original statement was too general and could be misleading. To address this, we revised the sentence to focus on the comparative evaluation carried out and explicitly linked it to the Med-CORDEX initiative. The revised text now reads:*
  *"This highlights the importance of thoroughly evaluating standalone routing models for their performance and limitations, before integrating them into regional coupled climate or Earth system models—such as the ENEA-REG atmosphere-land-river-ocean (ALRO) coupled model, developed within the framework of the Mediterranean CORDEX (Med-CORDEX) initiative."*

6) Line 84. This is the first time you mention Med-CORDEX. Even though you define CORDEX earlier, you assume that the reader is familiar with this particular experiment. Suggest adding more context for the reader. Maybe also point to Figure 1 here.

- *We thank the reviewer for this suggestion. In response, we added context about the Med-CORDEX initiative earlier in the manuscript (immediately after the revised line 75), explaining its aims and geographic domain, with the following text:*
*"Med-CORDEX aims to advance fully coupled regional climate simulations over the Euro-Mediterranean domain, which encompasses the Mediterranean and Black Seas and their contributing catchments (excluding the Nile), by improving the representation of key Earth system components, including atmospheric processes, land surface, hydrology and ocean dynamics (Ruti et al., 2016)."*
*Additionally, we now explicitly point to Figure 1 at the mention of the "Med-CORDEX domain" (line 84), so that the reader can better understand the spatial context.*

7) Line 91-93. The calibration seems like an add-on as written. Can you better integrate this into your experimental design? Why don't you do calibration experiments on CaMa-Flood?

- *We thank the reviewer for raising this point. This concern was also addressed in our response to major comment 2. In short, calibration was presented as a complementary "add-on" analysis because (i) CaMa-Flood and Noah-MP do not support basin-specific calibration in our framework, (ii) calibration of WRF-Hydro was facilitated thanks to the PyWRFHydroCalib package, and (iii) calibration is computationally expensive. For these reasons, we kept the baseline intercomparison restricted to non-calibrated configurations to ensure fairness, while presenting WRF-Hydro calibration separately to illustrate its added value.*

8) Line 95-96. Going forward, to show improvement, it's best to compare to an existing baseline (such as the Med-CORDEX simulations, or the underestimating HD model you talk about earlier).

- *We thank the reviewer for this suggestion. Following the same concern raised in the major comment on including a baseline, we have now integrated the HD model as a benchmark in Section 3 ("Results and Discussion"). Both qualitative (time series in Fig. S4) and quantitative (performance metrics in Tables S5 and S6) comparisons with HD are presented, highlighting its underestimation of discharge and providing a clear baseline against which CaMa-Flood and WRF-Hydro improvements can be assessed.*

9) Line 116: This paragraph doesn't fit as-is under the 2.1 heading (Study area and river discharge observations). If it does stay, then this is the first time the ENEA-REG model is mentioned, and the reader needs it to be defined, and more information about what it is. Looks like you don't say it's the atmosphere-land-ocean model until line 135 – should this information be moved to that section? Line 119: You mention the Med-CORDEX protocol – should some of the protocol info be mentioned in the introduction to help the reader with background?

- *We thank the reviewer for this observation. We chose to keep the paragraph under Section 2.1. To improve clarity, we have defined ENEA-REG at its first mention in the introduction as a coupled model together with MESMAR. Later in the introduction, before providing additional context on the Med-CORDEX initiative, we clarified that ENEA-REG is an atmosphere–land–river–ocean (ALRO) coupled model developed within the Med-CORDEX framework (see the reply to minor comment 5).*
*As suggested, information on the Med-CORDEX protocol has also been provided immediately after the context on the Med-CORDEX initiative:*
*"The Med-CORDEX Phase 3 protocol outlines common requirements for domain extent, spatial resolution, and coupling strategies—mandating, for example, a minimum resolution of 12 km for the atmosphere/land and 10 km for the ocean, and requiring river-to-ocean coupling. Further details on the protocol are available at https://zenodo.org/records/11659642.".*

10) Line 188: Why is the CaMa model not calibrated and compared during this study? Do you only calibrate wrf-hydro since the PyWrfHydroCalib package is available?

- *We thank the reviewer for this question. As noted in our reply to Major Comment 2 and Minor Comment 7, unlike WRF-Hydro, CaMa-Flood and Noah-MP do not currently support basin-specific or spatially variable parameter calibration, and Noah-MP (within ENEA-REG) is not two-way coupled with CaMa-Flood. In addition, calibration of WRF-Hydro was facilitated by the availability of the PyWRFHydroCalib package, whereas no equivalent tool exists for CaMa-Flood. For these reasons, only WRF-Hydro was calibrated and compared in this study.*

11) Figure 2, Figure 3, Table 2 etc: Is there a reason to not present all the results together? So include the three (i) default CaMa, (ii) default WRF-Hydro, and (iii) calib WRF-Hydro?

- *We thank the reviewer for this suggestion. They are not presented together for the fact that the calibration was considered as a separate section for the reasons mentioned previously (see reply to major comment 2, minor comment 7, and minor comment 10), so that the related results are presented in that section. Moreover, when integrating all the results together, some figures—specifically the Taylor diagram—became overloaded, making it difficult to read the results and distinguish the points.*

12) Line 310. You mentioned earlier that the HD model also tends to underestimate discharge. Is there a way to compare how much HD underestimates vs these models? It can help if there is a prior baseline to compare to, even if only qualitatively.

- *We agree with the reviewer's point. As part of the added baseline analysis (see reply to major comment 1), we now explicitly quantify the HD model underestimation relative to observations, using both 0.5-degree and 5-minute resolutions. These results (Section 3; Fig. S4; Tables S5 and S6) demonstrate that the underestimation is most pronounced in terms of variability and high-flow reproduction, and they serve as a direct benchmark against which WRF-Hydro and CaMa-Flood performance improvements can be evaluated.*

13) Line 434. This seems better to either go earlier (after Figure 5 and the KGE results) or in a discussion section, otherwise it seems out of place and hard to follow.

- *We thank the reviewer for the helpful suggestion. The line and related sentences have been moved to follow the discussion of Figure 5 and the KGE results.*

14) Line 464. This is a new topic and I recommend adding a new subsection here, and some introductory sentences to guide the reader on what was done and why. Also, I'm not following what is being shown in Table S5. Line 468 – I suggest reminding the reader that the ENEA-REG model is downscaled with WRF. The Table S5 caption needs to also remind the reader that WRF is downscaling the ENEA-REG. I had trouble crosswalking what was in the text and in the supplemental Table S5, suggest revising. Is there a better/clearer way to show the results in Table S5, and/or should it be brought into the main manuscript rather than the Supplemental?

- *We thank the reviewer for this valuable suggestion. Following the advice, we introduced a new subsection titled "3.2.2 Effects of calibration on runoff generation and partitioning", with an introductory paragraph to guide the reader on what was done and why:*
  *"To better understand the source of performance improvements following calibration, we investigated how the calibrated parameters influenced key hydrological processes—particularly runoff generation and the surface–subsurface flow partitioning. This analysis helps explain the hydrological changes driving model behaviour across the Mediterranean basins. Specifically, we examined simulated total runoff, as it serves as the primary input for CaMa-Flood, and the ratio of subsurface to total runoff. As a reminder, runoff is simulated by the Noah-MP land surface model, which is integrated within WRF-Hydro and embedded in WRF—used to dynamically downscale ERA5 data within the ENEA-REG coupled model."*
- *To better show the results previously in Table S5, we replaced the table with a new figure brought into the main manuscript. The figure consists of two subplots using grouped bar charts: subplot (a) represents average total runoff and subplot (b) represents the subsurface-to-total runoff ratio, with each group of bars corresponding to a river basin.*

[Figure]

**Figure 9: Comparison of a) average daily total runoff and b) the ratio of subsurface to total runoff across each river basin as simulated by ENEA-REG (orange), WRF-Hydro with default parameters (blue), and WRF-Hydro with calibrated parameters (green).**

- *Actually, the ENEA-REG model is not downscaled with WRF, but rather WRF is the atmospheric component of ENEA-REG, used to dynamically downscale ERA5 meteorological data. For this reason, and to avoid confusion, the new figure caption does not mention "WRF" but instead refers to ENEA-REG. At the same time, the introductory paragraph (above) includes a reminder explaining that WRF is used to downscale ERA5 data within ENEA-REG.*

15)   Line 493. I don't see a Fig S7.

- *We thank the reviewer for the observation. Figure S7 is included in the Supplement, immediately following Figure S6.*

[Figure]

| Basin | Mean Bias (%) |
|---|---|
| Danube | -1.2 |
| Rhone | 17.7 |
| Po | 1.7 |
| Ceyhan | 2.7 |
| Adige | 32.5 |
| Tiber | -27.3 |
| Maritsa | -9.9 |
| Goeksu | -14.3 |
| Arno | -9.5 |
| Kopru | 27.7 |
| Ebro | -5.2 |

**Figure S7: Distribution of the mean daily rainfall percent bias (100 x (WRF-EOBS)/EOBS)**

16) Line 495. Would be good to recognize that including a baseline for Euro-Med models would be helpful to see if there was an improvement.

- *We thank the reviewer for this helpful suggestion. The revised manuscript now explicitly incorporates the HD model as a baseline benchmark for Euro-Mediterranean simulations (Section 3; Fig. S4; Tables S5 and S6). This addition addresses the reviewer's point by providing a clear reference framework, making it possible to show that both CaMa-Flood and WRF-Hydro outperform the HD model.*

*References:*

*Cisterna-García, A., González-Vidal, A., Martínez-Ibarra, A., Ye, Y., Guillén-Teruel, A., Bernal-Escobedo, L., and Skarmeta, A. F.: Artificial intelligence for streamflow prediction in river basins: a use case in Mar Menor, Sci Rep, 15, 19481, https://doi.org/10.1038/s41598-025-04524-0, 2025.*

*Hagemann, S. and Dümenil, L.: A parametrization of the lateral waterflow for the global scale, Clim Dyn, 14, 17–31, https://doi.org/10.1007/s003820050205, 1997.*

*Stacke, T. and Hagemann, S.: HydroPy (v1.0): a new global hydrology model written in Python, Geosci Model Dev, 14, 7795–7816, https://doi.org/10.5194/gmd-14-7795-2021, 2021.*

*Ruti, P. M., Somot, S., Giorgi, F., Dubois, C., Flaounas, E., Obermann, A., Dell'Aquila, A., Pisacane, G., Harzallah, A., Lombardi, E., Ahrens, B., Akhtar, N., Alias, A., Arsouze, T., Aznar, R., Bastin, S., Bartholy, J., Béranger, K., Beuvier, J., Bouffies-Cloché, S., Brauch, J., Cabos, W., Calmanti, S., Calvet, J.-C., Carillo, A., Conte, D., Coppola, E., Djurdjevic, V., Drobinski, P., Elizalde-Arellano, A., Gaertner, M., Galàn, P., Gallardo, C., Gualdi, S., Goncalves, M., Jorba, O., Jordà, G., L'Heveder, B., Lebeaupin-Brossier, C., Li, L., Liguori, G., Lionello, P., Maciàs, D., Nabat, P., Önol, B., Raikovic, B., Ramage, K., Sevault, F., Sannino, G., Struglia, M. V, Sanna, A., Torma, C., and Vervatis, V.: Med-CORDEX Initiative for Mediterranean Climate Studies, Bull Am Meteorol Soc, 97, 1187–1208, https://doi.org/10.1175/BAMS-D-14-00176.1, 2016.*

---

## Author Comment (AC3)

**Response to Anonymous Referee #2**

The paper by Hamitouche et al. presents the performance of two hydrological models, namely CaMa-Flood and WRF-Hydro, driven by the ENEA-REG atmosphere-land-ocean coupled model run at a 12km resolution over the Med-Cordex domain. The performances analyzed concern default (uncalibrated) versions of both models, as well as a calibrated version of WRF-Hydro. While the authors highlight the higher performance of the calibrated version of WRF-Hydro, they present this work (to my understanding) as a preliminary step for further application "for offline atmosphere-hydrological simulations to close the water cycle at the land-ocean interface" (L526).

In brief, the key findings presented in this paper are the performances of two uncalibrated models and one calibrated model, which utilize data from an offline climate model as input, on a series of Mediterranean basins. These findings can be, of course, useful, but presented alone, rather than well inserted in a workflow providing much more advanced results, are no more than calibration exercises or, in the case of the uncalibrated models, simple applications of already existing hydrological models, not providing any novelty or relevant contribution to the field. Indeed, model calibration is a preliminary and unavoidable necessity for any hydrological model. The two research questions presented (LL95-97) have quite obvious answers, especially the second.

> - *We respectfully disagree with the reviewer's assessment that the study lacks novelty. The selection of an appropriate hydrological model is a critical step for accurately representing rivers and discharge simulations into global or regional Earth System Models. In this context, our work provides the first model-to-model comparison at the Mediterranean scale, offering an alternative to the pre-existing and less compatible HD model. This represents a key contribution by identifying solutions to improve river representation and freshwater fluxes into ocean models, which are central to coupled Earth system modelling efforts in the Med-CORDEX framework.*
>
> *Regarding the research questions, we acknowledge that their answers may appear generally intuitive. However, the results demonstrate that the extent of improvement following calibration is not trivial. For example, in three basins, the default parameters already yielded optimal performance, and calibration did not produce further improvement. This underscores that calibration outcomes are basin-dependent and not universally beneficial, which is an important and non-obvious finding for guiding future modelling applications in this region.*

I strongly suggest that the research presented be strengthened by associating this initial evaluation/calibration step with a more meaningful analysis (e.g., long-range hydrological reanalysis, sensitivity analysis, or exploring issues related to hydrological extremes). The results achieved so far seem too preliminary.

> - *We thank the reviewer for this suggestion. We agree that more advanced analyses— such as long-range hydrological reanalyses, sensitivity experiments, or the study of hydrological extremes—are highly relevant and form a natural continuation of this work. Indeed, we are currently developing a follow-up study that substitutes the HD model with the more advanced hydrological model assessed here, in order to investigate future projections of hydrological extremes over the Euro-Mediterranean*

*region. However, the scope of such an analysis is too broad to be integrated into the present manuscript.*

*The current study was designed as a preliminary and essential step: evaluating and calibrating hydrological models for their suitability within Euro-Mediterranean coupled modelling systems. This type of contribution is directly aligned with model evaluation papers in Geoscientific Model Development (GMD), which explicitly include the comparison of the performance of different model configurations or parameterizations as a recognized manuscript category (see GMD guidelines). We therefore consider the presented analysis to be sufficiently substantive for the scope of this journal, while laying the foundation for more advanced applications in subsequent work.*

Furthermore, in reviewing the study, I suggest considering the following comments.

1) Introduction: It is unclear why these two hydrological models were chosen and not (also) others. In addition, especially concerning WRF-Hydro, there is a vast amount of references describing significantly more advanced research, in which the calibration issue has been overcome with interesting strategies to be taken into consideration, simulating even multiple basins simultaneously for a large number of years, both in one-way and fully-coupled modes. A basic search on Scopus reveals approximately 300 documents containing "WRF-Hydro" in the Article Title, Abstract, or keywords.

*- We thank the reviewer for this observation. The choice of CaMa-Flood and WRF-Hydro is motivated by both scientific and technical reasons, including the modularity of the model which allows to easily couple them with atmospheric and ocean components. Besides, our selection builds on a previous study in which we analysed the impact of different Noah-MP runoff schemes on discharge simulations using CaMa-Flood. That study showed overall good performance against observations but also revealed important limitations, such as delays in capturing seasonal peak flows due to inherent constraints in CaMa-Flood. These limitations were successfully resolved by WRF-Hydro when tested in the same framework (Gochis et al., 2021), as stated in LL66–75. This provided a strong rationale for the present model-to-model comparison focusing on the most important Mediterranean rivers.*

*From a scientific perspective, both models are highly relevant for the Euro-Mediterranean regional coupled modelling framework. WRF-Hydro is the hydrological extension of WRF and uses Noah-MP as its land surface scheme. Since WRF and Noah-MP are the default atmospheric and land surface components of Med-CORDEX regional coupled models such as ENEA-REG, WRF-Hydro represents a natural candidate for improved river representation. CaMa-Flood, on the other hand, has been extensively validated for large-scale discharge simulations and provides a robust benchmark for comparison, as stated in LL82–85.*

*Regarding calibration, our objective was not to identify the best calibration strategy, but rather to assess the extent to which calibration can improve discharge simulations in this regional framework. While advanced calibration studies on WRF-Hydro exist, these are often constrained to smaller basins, limited numbers of parameters, or short*

*simulation periods due to computational demands. For example, Sofokleous et al. (2023) calibrated 31 small mountainous catchments (5–115 km²) in Cyprus, while Tijerina-Kreuzer et al. (2021) compared WRF-Hydro and ParFlow over the CONUS domain without calibration, simulating only one water year at 1 km resolution at the cost of a full year of supercomputing. In our context, the basins of interest are much larger, reaching hundreds of thousands of square kilometres, and many Mediterranean sub-catchments remain ungauged, making detailed sub-basin calibration infeasible. In addition, we would underline that this study represents a first attempt to use WRF-Hydro over the whole continental Europe.*

*Finally, we acknowledge promising recent approaches such as the iterative Ensemble Smoother (iES) applied to WRF-Hydro by RafieeiNasab et al. (2025), which provides ensemble calibration with uncertainty bounds. While powerful, such methods are computationally very expensive and beyond the scope of the present study. Our focus is therefore on demonstrating the potential of calibration within a tractable and regionally relevant setup.*

2) One model requires daily runoff as input, while the other uses 6-hour meteorological data. One model is calibrated, the other not. Any comparison between these two models looks quite unbalanced. I would propose that the results of the analysis be presented as a sequence of steps to further improve the hydrological output (even though I don't consider the uncalibrated analysis interesting).

- *We thank the reviewer for this important observation. To ensure fairness in the comparison, we treated the WRF-Hydro calibration as a separate analysis, presented in its own subsection. This structure avoids mixing calibrated and uncalibrated results and makes the rationale for the calibration exercise clearer, as reflected in the revised section headings and flow of the paper.*

  *Regarding model inputs, we clarify that CaMa-Flood and WRF-Hydro operate differently by design. CaMa-Flood is a hydrodynamical model that only relies on runoff input, whereas WRF-Hydro is a full hydrological framework embedding Noah-MP to simulate runoff and route it to discharge. Within ENEA-REG, CaMa-Flood is coupled to WRF, where Noah-MP generates runoff from the same 6-hourly meteorological forcing used by WRF-Hydro. This ensures that both models rely on Noah-MP runoff driven by the same atmospheric inputs, thereby making the comparison consistent.*

3) The calibration and validation methodology is rather unclear. Reading LL231-235, the reader cannot understand for each basin analysed how many years were used for calibration and how many for validation. The sentence "The optimized parameters from the calibration were then used to evaluate the model over the entire 1990–2014 period" suggests that the years used for calibration were also used for validation. A simple Table would have helped. No hydrographs are shown in the entire manuscript, nor in the Supplementary Material. In addition, a detailed comparison between ENEA-REG and WRF-Hydro runoff is lacking (I can only see Fig. S4, which shows the mean seasonal cycle for only one river).

- *We thank the reviewer for pointing this out. The calibration was performed for 5 years (as stated in L229), plus a one-year spin-up at each iteration (L230). The specific calibration periods varied across basins depending on the availability of continuous observations. Validation was then carried out over the entire available period within 1990–2014 (excluding missing values), using the calibrated parameters. For example, in the Danube basin, the calibration period was 01/10/1999–01/10/2005 (including one-year spin-up), while the evaluation extended over 1990–2014.*

*To clarify this aspect, we added a new Supplementary Table (Table S4) reporting the spin-up and calibration periods for each basin, and we revised LL231–234 accordingly.*

*Revised LL231–234:*
*"The specific spin-up and calibration periods varied across basins and were selected based on the availability of reliable and continuous observational records (i.e. at least 5-year all falling within the 1990–2014 period), and are provided in Table S4 in the supplement. The optimized parameters from the calibration were then used to evaluate the model over the entire available period within 1990–2014, focusing on ensuring consistent performance across the selected basins."*

*Table S1: Summary table of spinu-up and calibrations periods for each river basin*

| Basin | Spin-up period | Calibration period (including one-year spin-up) |
|---|---|---|
| Danube | 01/10/1994 – 01/10/1999 | 01/10/1999 – 01/10/2005 |
| Rhone | 01/10/1985 – 01/10/1990 | 01/10/1990 – 01/10/1996 |
| Po | 01/10/2000 – 01/10/2005 | 01/10/2005 – 01/10/2011 |
| Ceyhan | 01/10/1995 – 01/10/2000 | 01/10/2000 – 01/10/2006 |
| Adige | 01/10/2000 – 01/10/2005 | 01/10/2005 – 01/10/2011 |
| Tiber | 01/10/1992 – 01/10/1997 | 01/10/1997 – 01/10/2003 |
| Maritsa | 01/10/1994 – 01/10/1999 | 01/10/1999 – 01/10/2005 |
| Goeksu | 01/10/1995 – 01/10/2000 | 01/10/2000 – 01/10/2006 |
| Arno | 01/10/2000 – 01/10/2005 | 01/10/2005 – 01/10/2011 |
| Kopru | 01/10/1995 – 01/10/2000 | 01/10/2000 – 01/10/2006 |
| Ebro | 01/10/1994 – 01/10/1999 | 01/10/1999 – 01/10/2005 |

*Additionally, discharge hydrographs have been included in the Supplement (as Fig. S4), showing both CaMa-Flood and WRF-Hydro simulations compared to observations and to the HD model (as a benchmark).*

[Figure]

*Figure S4: Observed and simulated daily discharge for the Ebro, Rhone, Tiber, Goeksu and Po rivers for common 10 years from 1995 to 2004. Simulations include ENEA-REG–driven WRF-Hydro and CaMa-Flood, as well as the HD model at 0.5° and 5-minute spatial resolutions, evaluated near the corresponding gauge stations.*

[Figure]

*Figure S4 (continuity): Observed and simulated daily discharge for the Kopru, Maritsa, Ceyhan and Danube rivers for common 10 years from 1999 to 2008. Simulations include ENEA-REG–driven WRF-Hydro and CaMa-Flood, as well as the HD model at 0.5° and 5-minute spatial resolutions, evaluated near the corresponding gauge stations.*

[Figure]

*Figure S4 (continuity): Observed and simulated daily discharge for the Adige and Arno rivers for common 10 years from 2005 to 2014. Simulations include ENEA-REG–driven WRF-Hydro and CaMa-Flood, as well as the HD model at 0.5° and 5-minute spatial resolutions, evaluated near the corresponding gauge stations.*

- *Finally, the comparison of ENEA-REG and WRF-Hydro runoff was previously presented in Supplementary Table S5 and discussed in LL466–489. Now it has been revised and is provided in a dedicated subsection ("3.2.2 Effects of calibration on runoff generation and partitioning"). Instead of Supplementary Table S5, we now include a clearer grouped bar chart (as Fig. 9 in the main manuscript), consisting of two subplots: (a) average total runoff and (b) subsurface-to-total runoff ratio, with each group of bars corresponding to a river basin.*

[Figure]

*Figure 9: Comparison of a) average daily total runoff and b) the ratio of subsurface to total runoff across each river basin as simulated by ENEA-REG (orange), WRF-Hydro with default parameters (blue), and WRF-Hydro with calibrated parameters (green).*

4) In addition, concerning the calibration strategy, the values of the calibrated parameters should be duly analysed and discussed. For example, the parameter smcmax represents the maximum soil moisture content for each soil type. Why is it higher than 1 in most basins? Please consider the issue of equifinality seriously while dealing with a multi-parameter calibration.

- *We thank the reviewer for this comment. The values of the calibrated parameters, including smcmax, and their influence on hydrological processes are already analysed and discussed in detail in the Supplement (Sect. S1 (LL15-27), Table S4). As stated there, smcmax is a scalar multiplier of soil porosity with a valid range of 0.8–1.2, so values above 1 remain consistent with the model formulation and correspond to increased infiltration and soil evaporation. In addition, our discussion explicitly addresses the issue of equifinality, noting that parameter interactions can compensate or dominate each other's effects, ultimately determining the hydrological outcome.*

*LL15-27 in the supplement:*

*"The calibration process revealed some notable overall trends across all basins. There*

*was a consistent increase in refkdt, which controls runoff partitioning and results in a higher proportion of subsurface runoff compared to surface runoff. Additionally, the slope parameter consistently decreased, reducing deep drainage from the soil column to the groundwater reservoir. The zmax parameter showed a general increase across most basins, except for Göksu, leading to reduced baseflow contributions to river discharge.*

*While these three parameters exhibited relatively uniform behaviour across all basins, others displayed more variable trends. For instance, parameters such as bexp, smcmax, dksat, and rsurfexp showed both increases and decreases depending on the basin, reflecting local hydrological conditions.*

*This variability underscores the complexity of the calibration process. The interplay between parameters often determines the final hydrological outcome, as the effect of one parameter can be moderated or even compensated by another influencing the same physical process. In some cases, one parameter may dominate over others, masking their effects. These findings highlight the importance of considering the combined influence of multiple parameters to ensure accurate and balanced calibration for hydrological modelling.”*

5) L146: None of the simulations considers reservoir operations and lakes. That's probably one reason why some of the rivers are simulated very poorly. The rivers' flow should be preliminary naturalized.

- *We thank the reviewer for raising this important point. Indeed, none of the simulations include reservoir operations or lakes. This decision was taken primarily because of the lack of consistent and comprehensive information on reservoir management and characteristics across all Mediterranean basins, which span multiple countries and governance levels. To ensure a fair spatial evaluation across the study domain, we chose not to incorporate reservoirs. We agree that this is a limitation and acknowledge that the absence of reservoir regulation is likely one reason for poor performance in some rivers.*

  *Revised LL146-147:*

  *“Both CaMa-Flood and WRF-Hydro hydrological models are run for the period 1990-2014, after five years of spin-up. None of the simulations considers reservoir operations and lakes, due to the lack of consistent and comprehensive information on reservoir management and characteristics across all Mediterranean basins. This choice ensures a fair spatial evaluation across the study domain, but we acknowledge it as a limitation that may contribute to reduced performance in some rivers.”*

6) L178: Hydrological routing model components of WRF-Hydro were run on the same 6 km spatial resolution grid. Such a resolution is very low for most of the Mediterranean catchments. Additionally, this approach does not utilize one of the main features of WRF-Hydro, namely subgrid disaggregation and aggregation.

- *We agree with the reviewer that a 6 km spatial resolution is coarse from a hydrological perspective and does not exploit one of the main features of WRF-Hydro, namely subgrid disaggregation and aggregation. However, in the context of Euro-Mediterranean coupled models, even coarser resolutions are typically used (e.g., 0.5° ≈ 55 km or 5' ≈ 11 km). Thus, the 6 km grid still represents a step forward in improving river representation and discharge simulations. While higher resolutions are certainly preferable, they are strongly constrained by the availability of computational resources. For our purposes, the chosen resolution was a compromise that allowed us to meet the objective of closing the water cycle at the land–ocean interface across the regional scale.*

7) L200-202: "snowmelt parameters were left at their default settings due to the limited relevance of snowmelt in most of the study region and for potential future regionalization over other snow-free basins." This sentence sounds very strange for basins for which the Alps or the Pyrenees are fundamental. Probably, this is another reason why some of the rivers are simulated very poorly.

- *We thank the reviewer for raising this important point. Snowmelt processes indeed play a significant role in several Mediterranean basins, particularly the Ebro, Po, Rhone, and Danube. In this study, snowmelt parameters were left at their default settings to ensure consistency across basins and fairness in the intercomparison, and for possible regionalization over other snow-free basins. We acknowledge that this choice may have limited performance in snow-influenced basins. However, it is worth noting that even without explicit calibration of snow-related parameter, the performance in these basins was satisfactory, with Kling–Gupta Efficiency (KGE) scores improving from intermediate to good after calibration (e.g., Danube: 0.46 → 0.66; Rhone: 0.54 → 0.65; Ebro: 0.46 → 0.65). This suggests that while snow processes are important, the calibrated parameters affecting other hydrological processes (e.g., infiltration, runoff partitioning) also played a major role in improving performance. We now explicitly acknowledge in the text that future work should investigate snowmelt parameter calibration to further refine simulations in these catchments.*

  *Revised LL199-202:*
  *"In this study, calibration focused on the first four groups, covering 16 parameters (Table S3), while snowmelt parameters were left at their default settings to ensure consistency across basins and fairness in the intercomparison, and for potential future regionalization over other snow-free basins. We acknowledge that this choice may limit performance in snow-influenced basins and that future work could further refine results by including these parameters."*

8) Conclusions: Regarding calibration for fully coupled atmosphere-hydrological models, please note that many papers demonstrate that it is based on different concepts compared to offline calibration, and it should not be directly based on observed meteorological inputs.

- *We thank the reviewer for this comment. Following your suggestion, we have revised the text to remove the phrase "based on observed meteorological inputs" and now refer only to a tailored calibration approach.*

*References:*

*Gochis, D. J., Barlage, M., Cabell, R., Casali, M., Dugger, A., Fitzgerald, K., Mcallister, M., McCreight, J., Rafieeinasab, A., Read, L., Sampson, K., Yates, D., and Zhang, Y.: The WRF-Hydro Modeling System Technical Description. (Version 5.2), NCAR Technical Note, 108 pp, https://ral.ucar.edu/sites/default/files/docs/water/wrf-hydro-v511-technical-description.pdf (last access: 27 February 2025), 2021.*

*Sofokleous, I., Bruggeman, A., Camera, C., and Eliades, M.: Grid-based calibration of the WRF-Hydro with Noah-MP model with improved groundwater and transpiration process equations, J Hydrol (Amst), 617, 128991, https://doi.org/10.1016/j.jhydrol.2022.128991, 2023.*

*Tijerina-Kreuzer, D., Condon, L., FitzGerald, K., Dugger, A., O'Neill, M. M., Sampson, K., Gochis, D., and Maxwell, R.: Continental Hydrologic Intercomparison Project, Phase 1: A Large-Scale Hydrologic Model Comparison Over the Continental United States, Water Resour Res, 57, e2020WR028931, https://doi.org/10.1029/2020WR028931, 2021.*

*RafieeiNasab, A., Fienen, M. N., Omani, N., Srivastava, I., and Dugger, A. L.: Ensemble Methods for Parameter Estimation of WRF-Hydro, Water Resour Res, 61, e2024WR038048, https://doi.org/10.1029/2024WR038048, 2025.*